# Design and Performance of X-Band SAR Payload for 80 kg Class Flat-Panel-Type Microsatellite Based on Active Phased Array Antenna

Seok Kim [1], Chan-Mi Song [1], Seung-Hun Lee [1], Sung-Chan Song [1] and Hyun-Ung Oh [2,*]

1    Hanwha Systems, Inc., Aerospace Systems R&D Center, Yongin 17121, Korea;
     seok0511.kim@hanwha.com (S.K.); cm.song@hanwha.com (C.-M.S.); shlee3970@hanwha.com (S.-H.L.);
     sungchan.song@hanwha.com (S.-C.S.)
2    Space Technology Synthesis Laboratory, Department of Smart Vehicle System Engineering,
     Chosun University (Agency for Defense Development: Additional Post), 375 Seosuk-Dong, Dong-Gu,
     Gwangju 61452, Korea
*    Correspondence: ohu129@chosun.ac.kr

**Abstract:** The small synthetic aperture radar (SAR) technology experimental project (S-STEP) mission aims to develop an innovative spaceborne SAR microsatellite as a constellation of 32 microsatellites featuring a high-resolution stripmap mode of 1 m. The S-STEP is a spaceborne SAR microsatellite technology demonstration program in which innovative approaches have been proposed and investigated for SAR payload system designs for improving the development speed, affordability, size and weight parameters, and quality of SAR satellite systems. In this study, the major design approach includes a bus–payload integrated flat-panel-type SAR payload based on an active phased-array antenna. This study conducted an SAR image performance analysis considering the mission requirements to validate the feasibility of the innovative SAR payload design of the S-STEP. These performance analysis results are presented to demonstrate the effectiveness of the proposed SAR payload design approach under the new space paradigm.

**Keywords:** microsatellite; new space; synthetic aperture radar (SAR); X-band; small satellite

## 1. Introduction

The spaceborne synthetic aperture radar (SAR) has been successfully used to systematically and continuously monitor the dynamic evolution of the geosphere, biosphere, hydrosphere, and cryosphere of the Earth's surface in scientific, civil, commercial, and military applications. This is because the SAR can provide high-resolution images for remote sensing applications regardless of sunlight illumination and weather conditions [1–4].

In military applications, the maximum possible number of images of a particular area of interest (AoI) are required to be captured in a short time period according to the established mission plan to improve the surveillance and reconnaissance mission performances at national borders and military facilities [5,6]. In addition, the observe–orient–decide–act (OODA) loop model is predominantly used to establish a command and control (C2) system for rapid response and operation against arbitrary threats, especially time-critical or time-sensitive targets, such as the transporter–erector–launcher (TEL) [7]. The modified or extended OODA loop comprises inner and outer loops [8], wherein the inner loop cycle should be minimized for timely information acquisition. Accordingly, a microsatellite SAR system with a fast-revisit interval has been recently introduced and widely reviewed.

Based on the "Faster, Better, and Cheaper (FBC)" strategy [9] promoted by NASA since 1992, numerous new space companies have accelerated their innovation under the umbrella of the "New Space Paradigm" (NSP). As a representative example of these innovations, the

spaceborne SAR is actively undergoing industrial innovation owing to its aforementioned advantages. The emergence of the NSP has revolutionized the development philosophy and methodology of the global space industry [10]. NSP, which is also called Space 4.0 or Space Renaissance, is a collective reference expression of a methodology that boldly breaks through the existing government-led "Old Space" development philosophy. Instead, it leads to the rapid development of new space systems by adopting agile approaches and exploiting the latest commercial-off-the-shelf (COTS) technologies. The introduction of concepts such as "Smaller" and "Lighter" to the FBC development strategy, especially in the field of small satellites, has accelerated the development and implementation of several innovative space development programs under the NSP. A key area of application of the NSP is in the small satellite constellation for providing services such as surveillance and reconnaissance or communications over a large area, including the world. In addition, the shortening of the development period and reduction in the cost through large-scale mass production using standardized satellite platforms, innovative manufacturing methods such as additive 3D printing techniques, and automated test processes yield small satellite constellations for diverse challenging missions. Such missions include SAR imaging with a short revisiting period, near-real-time or real-time remote sensing, global internet services, and low-latency high-speed communications. Thus, owing to its several advantages, small SAR satellite constellations can complement or replace existing medium/large SAR satellite missions such as Sentinel-1, TerraSAR-X [11,12], and KOMPSAT-5 [13].

A graphical comparison of the SAR image resolutions in terms of the mass of the microsatellites being developed recently is shown in Figure 1 [14]. As indicated in the figure, these microsatellite SAR constellations, such as ICEYE and Capella SAR, can provide cost-effective and practical SAR images with a resolution comparable to those of medium/large SAR satellites such as KOMPSAT-5. Developed by ICEYE in Finland and launched in 2018, ICEYE X2 (92 kg) is known as the world's first commercial SAR in the microsatellite class based on its proof-of-concept model, ICEYE X1 [15,16]. It is the most outstanding representative microsatellite under the NSP that captures SAR images using stripmap, spotlight, and ScanSAR modes. These are based on the X-band active phased-array SAR antenna, which is a $3.2 \times 0.4$ m microstrip patch-array antenna with 320 transmit/receive modules (TRM). The Capella SAR, developed by Capella Space in the United States, is a 107 kg microsatellite SAR (Sequoia) that was successfully launched in 2020 [17–20]. In particular, this satellite features an X-band SAR payload that acquires spotlight-mode SAR images with a ground resolution of 0.5 m and uses a deployable tension–truss mesh reflector antenna with a 3.6 m diameter, similar to the AstroMesh [21] of the L-band SMAP SAR satellite or Oxford Space Systems [22], to minimize transmission power. This feature enables an outstanding continuous SAR image acquisition period of at least 9 min. The effective area of the Capella SAR antenna is 8 m$^2$ compared with the 1.3 m$^2$ antenna of ICEYE. Therefore, the Capella SAR antenna is more advantageous for obtaining higher SAR image quality, such as noise equivalent sigma zero (NESZ), than the ICEYE antenna. StriX-$\alpha$ is an X-band SAR microsatellite constellation containing 25 satellites, planned by Synspective in Japan, based on the development results of a prototype called MicroXSAR [23]. It is a 130 kg microsatellite that acquires SAR images with a ground resolution of 1–3 m. Specifically, Strix-$\alpha$ features a deployable passive phased-array antenna with $4.9 \times 0.7$ m honeycomb panels of a slotted waveguide array containing solar panels mounted on its rear. Furthermore, QPS-SAR is an X-band SAR microsatellite constellation developed by iQPS in Japan, and two launches have been completed in 2019 and 2021, respectively [24]. Similar to the Capella SAR, a total of 36 launches are planned for the QPS-SAR. These satellites use a deployable wrapped-rib parabolic mesh reflector antenna. This satellite weighs 100 kg with a 3.6 m diameter antenna that provides a high resolution of 1 m in spotlight mode and a standard resolution of 1.8 m in stripmap mode.

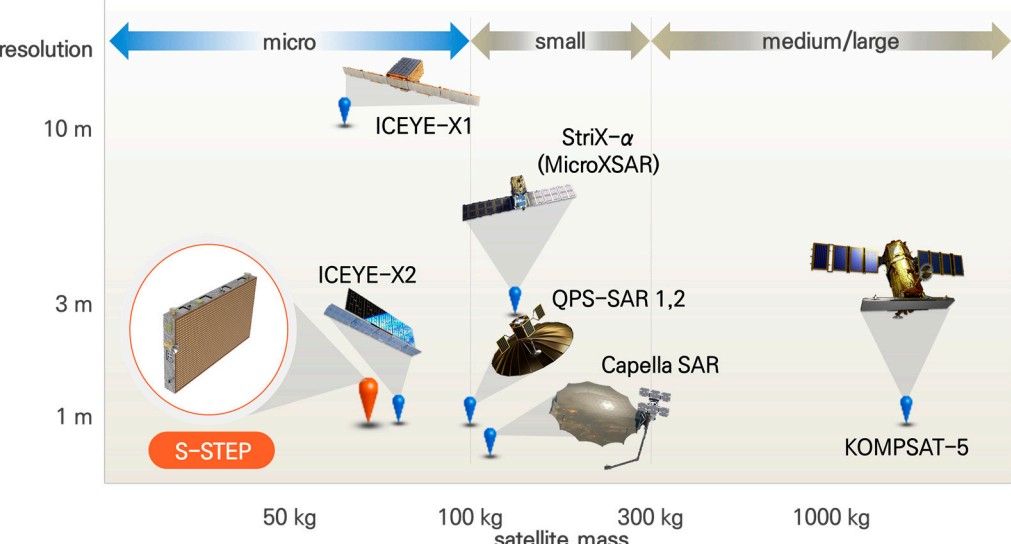

**Figure 1.** Comparison of satellite masses vs. SAR image resolutions.

The S-STEP system has adopted various challenging strategies to actively pursue a new space approach that is faster, cheaper, smaller, and lighter than the old space approach. The fundamental strategies are a flat-panel-type satellite structure integrated with both a bus and payload, a multifunctional mechanical structure of TRM, and a dedicated vibration-free orbital deployer (VFOD) [25,26]. As well as these strategies, the key design goal in terms of an SAR payload is to develop a lightweight SAR payload that can be applied to an 80 kg class microsatellite. In particular, the S-STEP mission aims to develop a multimode SAR payload system that can be mounted on a microsatellite. This payload is designed to provide various SAR image qualities in multiple modes, e.g., stripmap and ScanSAR modes, by utilizing antenna beam scheduling. Moreover, by integrating an SAR antenna with the satellite structure, the satellite mass can be reduced to achieve an 80 kg class microsatellite. Furthermore, a microstrip patch antenna is applied for lightweight miniaturization and cost reduction. An aperture-coupled structure that can produce a high antenna gain by increasing the antenna efficiency was applied to the array radiating elements, and a substrate integrated waveguide (SIW) cavity-backed microstrip patch antenna was applied instead of the existing metal cavity-back technique to enhance its manufacturability and antenna gain performance.

Essentially, the S-STEP SAR payload can provide a high resolution of 1 m in the stripmap mode, for which the resolution usually corresponds to that in the spotlight mode of medium or large SAR satellites. The spotlight mode SAR images exhibit limited coverage; however, the stripmap mode can acquire SAR images with continuous long coverage, which is advantageous for land-based security applications. Therefore, an antenna length of ~2 m was selected in the S-STEP to obtain images with a high resolution of 1 m in the stripmap mode.

This study proposes a design and performance analysis for a novel SAR payload suitable for an 80 kg class SAR microsatellite. The essential SAR payload design for the flat-panel-type satellites integrated with both bus and payload mechanical structures demands high-gain antennas, high-power TRMs, and miniaturized/lightweight unit designs to enable the SAR image acquisition for the required mission performance. A lightweight SIW cavity-backed microstrip patch antenna is used. A brick-type TRM is applied with a high-power GaN-type monolithic microwave integrated chip (MMIC) rather than a GaAs MMIC. An effective heat dissipation and structural reinforcement functions are employed to TRM assembly. More specifically, the multifunctional integrated structure of the TRM assembly enables effective heat dissipation through its integrated radiators when a large amount of heat is generated from TRMs during SAR imaging [26]. The structural stiffness of the satellite structure is reinforced by mechanically integrating the TRM housing with

the satellite body. Additionally, a transceiver capable of transmit (Tx)-band stretching by quadruple frequency multiplication was applied. Furthermore, a digital control unit based on a high-performance field-programmable gate array (FPGA) provides multiple functions of generating a wideband Tx waveform based on parallelized direct digital synthesis (PDDS), which is robust in the radiation environment in space, and digital processing of the received signal. These strategies contribute to minimize the satellite mass and facilitate the implementation of "lighter" SAR satellites. Moreover, a performance analysis was conducted on the S-STEP SAR payload to verify the feasibility of the proposed design. In this study, the proposal focuses more on the design and development of the SAR payload, rather than at the satellite level.

The remainder of this paper is organized as follows: The mission design and satellite system of the S-STEP are explained in Section 2. The SAR payload design proposed for the innovative S-STEP satellite is presented in Section 3, and the unit design of the SAR payload is briefly described in Section 4. Thereafter, the antenna model of the SAR payload, performance analysis results, and the conclusions of this study, as well as the future scope of this study, are discussed in Sections 5–7, respectively. The mechanical and thermal design strategies of the S-STEP are described in prior studies on S-STEP missions [25,26].

## 2. Mission and Satellite Design

The S-STEP is defined as a spaceborne SAR microsatellite technology demonstration program by developing an engineering qualification model of the SAR payload. The fundamental mission objective of the S-STEP is the continuous near-real-time monitoring of the Earth's surface using high-resolution SAR images through an affordably developed 80 kg class microsatellite SAR constellation. This constellation operates within an average revisit cycle of half an hour. These images can be employed in civil, scientific, military, and commercial applications. In military surveillance, these images are used to detect time-critical targets such as TELs, and in maritime surveillance, these are used to detect illegal fishing vessels (dark vessels). The S-STEP mission aims to develop a faster, better, smaller, and lighter 80 kg class SAR satellite in accordance with the NSP.

### 2.1. Overview

The satellites for the S-STEP mission involve 80 kg class X-band SAR microsatellite constellations based on an active phased-array antenna. In particular, there are six major mission requirements for the S-STEP, and the design lifetime of the mission is at least three years with a total mass of 80 kg. The mission involves two distinct SAR modes: a stripmap mode and 3-beam ScanSAR mode as showing Figure 2. Accordingly, the high-resolution mode acquires the SAR raw data with a 5 km swath with a ground resolution greater than $1 \times 1$ m (single look) using the stripmap technique. Comparatively, the wide-swath mode acquires the SAR raw data with a 15 km swath with a ground resolution at greater than $4 \times 4$ m (single look) using the ScanSAR technique. The continuous SAR image acquisition period is greater than 60 s for both imaging modes. Moreover, the raw data are transmitted to the ground segment in near-real-time via the X-band data link of the platform.

The key factors that influence the micro-SAR satellite performance are the average/maximum revisit cycle, response period, and coverage. Although more performance factors can be considered, the S-STEP mission derives the optimal satellite orbit and system parameters based on these key factors.

The design of the constellation is vital for satisfying the major mission requirements, such as the lifetime, revisit time, and SAR image quality. The constellation orbit significantly impacts the mission performance because multiple satellites perform a single mission. Additionally, the constellation of the S-STEP mission is designed by adopting the Walker–Delta technique that considers the number of launches, number of satellites, probability of collision, and the revisit time to the area of interest based on the mission concept and requirements.

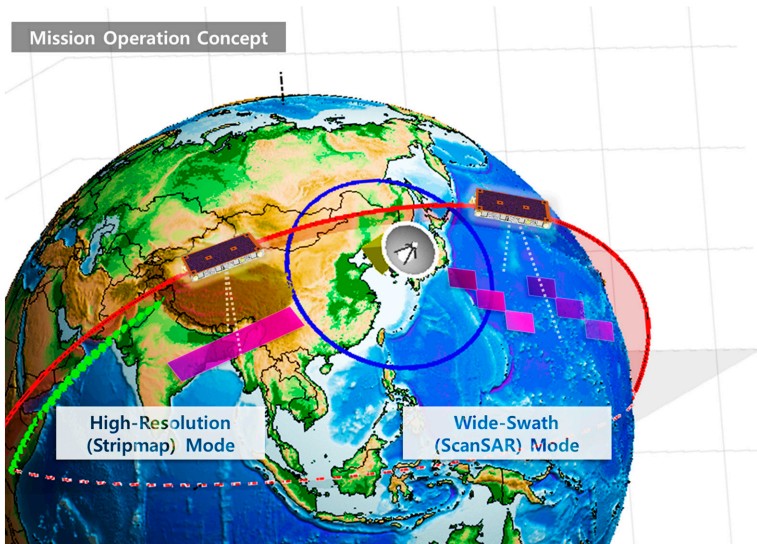

**Figure 2.** Operating concept of the S-STEP mission in the high-resolution and wide-swath modes.

The frozen orbit and repeat ground track (RGT) orbit are considered during the design of the satellite constellation orbits to enhance the mission plan efficiency and usability of the SAR images. In this study, the RGT orbit was designed with an altitude range of $510 \pm 10$ km and repeated every 14 d to facilitate mission planning.

To minimize the revisit time for the region of interest up to ~30 min, the orbit incidence angle was selected according to the longitude of the region of interest and subsequently optimized. This was because the revisit time of the satellite for a certain area depends on the orbit inclination. Therefore, the final constellation and orbit of the S-STEP mission were selected using multidisciplinary design optimization with FreeFlyer, commercial application software for satellite mission analysis, design, and operations based on the essential performance factors. The constellation of the microsatellite with a Walker–Delta pattern is operated on four orbit planes, and eight satellites are equally spaced on each plane to operate a total of 32 satellites. The mission orbit is nominally circular, non-sun-synchronous, and inclined. More specifically, the nominal altitude, orbits per day, repeat period, and inclination of the mission orbit are 510 km, 15 2/11, 13.75 d, and 44.2°, respectively [27]. A comparison of various satellite constellations based on their individual performance indices is illustrated in Figure 3 [28].

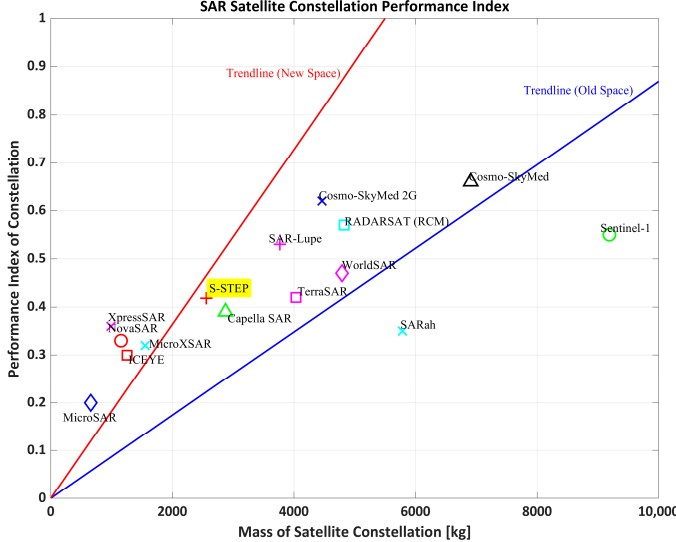

**Figure 3.** Comparison of the SAR performance indices as functions of satellite constellation masses.

### 2.2. Mission and Satellite

The S-STEP space segment was optimized for operation in low Earth orbit (LEO) with the required radar performance and technology capability maintained, and it is compatible with a wide range of launchers owing to its compact shape and low mass [27]. The flat-panel shape of the spacecraft efficiently fits the shape of typical launcher faring. The solar array and SAR antenna are displayed in Figure 4a,b, respectively. The majority of the electronic units are accommodated on the rear of the SAR antenna, the length and width of which are 1970 and 1060 mm, respectively. In the nominal altitude of the mission orbit, the SAR antenna off-nadir angle was 23.7°.

The critical tasks of the S-STEP satellite bus include providing the SAR instrument with adequate energy and precise pointing, as well as maintaining a stable orbit and appropriate thermal conditions. In addition, the spacecraft employs a fully passive thermal-control approach [26].

The system specifications of the S-STEP satellite are listed in Table 1. The S-STEP microsatellite bus is a low-cost platform that mostly adopts the COTS units in addition to the integrated avionics unit (IAU), attitude control system unit (ACSU), and X-band modulator (X-MOD). The onboard computer of the microsatellite SAR platform was developed as a unit in which the common components of the command and data handling unit, solid-state recorder unit, ACSU, and X-MOD comprising the platform were integrated and shared to form the IAU. The common components include a high-performance FPGA (Xilinx Kintex UltraScale), large-volume memory, and regulators. The IAU onboard computer includes a processor module based on the LEON3 Fault-Tolerant SPARC V8 soft-core processor, which is executed on the FPGA with RTEMS as a real-time operating system.

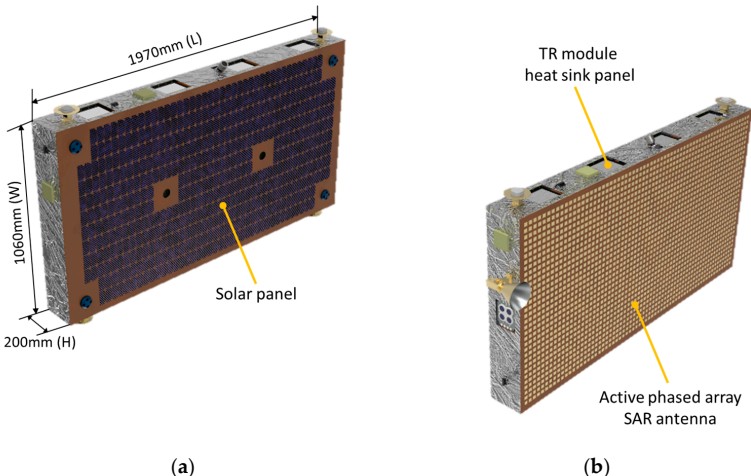

(**a**)                                                          (**b**)

**Figure 4.** Mechanical configuration of the S-STEP satellite viewed from the (**a**) solar-panel side and (**b**) SAR-antenna-panel side. Patches appearing in the SAR antenna in (**b**) are schematic representations for understanding. In an actual SAR antenna, the applied inverted patches are invisible.

**Table 1.** S-STEP satellite system specifications [25–27].

| Specification | Value |
| --- | --- |
| Mission lifetime | 3 years |
| Total spacecraft mass | 80.3 kg excluding launch adapter |
| Satellite size | 1970 × 1060 × 200 mm |
| Power | Generation: 340 W (BoL) Storage: 648 Wh |
| Inter-satellite link | RF (S-band, 20 kbps) |
| Telemetry telecommand (TMTC) | S-band (Up: 32 kbps; Down: 1 Mbps) |
| Datalink | X-band (1 Gbps) |
| Stabilization method | 3-axis stabilized |
| Pointing accuracy | 0.085° (3σ) |
| Image acquisition time | 60 s max |

*2.3. Imaging Modes*

The imaging modes of the S-STEP SAR offer operational modes that utilize the advantages of an active phased-array antenna, and these operational modes are simple for a small satellite and demonstration mission of the S-STEP. In particular, the S-STEP SAR can be operated in multiple modes, including stripmap and ScanSAR, to provide SAR images with varying resolutions, swath widths, or sensitivities for diverse applications [29]. Subsequently, the antenna beam scheduling is adjusted to operate the stripmap and ScanSAR modes based on the requirements of the SAR image quality. The essential performance parameters of the S-STEP SAR payload for high-resolution and wide-mode SAR image quality are listed in Table 2.

**Table 2.** S-STEP SAR payload key performance parameters.

| Parameters | High-Resolution Mode | Wide-Swath Mode |
| --- | --- | --- |
| Access region | 15°~35° | 15°~35° |
| Swath width | ≥5 km | ≥15 km |
| Resolution | ≤1 m @ 25° | ≤4 m @ 25° |
| NESZ | ≤−14 dB @ 25° | ≤−16 dB @ 25° |
| Peak sidelobe ratio | ≤−17 dB | ≤−17 dB |
| Integrated sidelobe ratio | ≤−12 dB | ≤−12 dB |
| Range ambiguity ratio | ≤−17 dB | ≤−17 dB |
| Azimuth ambiguity ratio | ≤−17 dB | ≤−17 dB |

2.3.1. High-Resolution Mode

Stripmap imaging is the most fundamental SAR technique, and the stripmap mode provides the finest cross-resolution that can be achieved by continuously acquiring SAR images without interruption. Unlike alternative SAR imaging techniques, angular steering is not performed with respect to the transmitted and received beams during image acquisition. Moreover, the antenna beam is fixed in the broadside direction. A schematic of the SAR operation in the stripmap mode is illustrated on the left-hand side of Figure 2. In S-STEP SAR, the stripmap mode is operated at a high resolution of 1 m, and it is designed to combine the advantages of the stripmap and spotlight modes. Thus, the acquisition of continuous high-resolution SAR images is possible in a long strip by selecting the antenna size with a high resolution of 1 m in the stripmap mode. This delivers an effective mission performance such as faster target detection and identification for surveillance applications. In general, the high-resolution mode can also be operated in the traditional stripmap imaging technique. The main advantage of an S-STEP SAR payload is that it can achieve a resolution of 1 m in the stripmap mode with high-beam agility owing to its full elevation steering capability. This is possible because of the extremely short antenna length in the along-track direction and the presence of one phase center per element in the elevation for the stripmap mode operation. This is extremely attractive, especially for military, surveillance, and security applications. The representative examples of these SAR satellite systems include Excubitor and Nimbus [30].

2.3.2. Wide-Swath Mode

The wide-swath mode operates the ScanSAR imaging technique in the burst or scanning mode and uses beam steering in the elevation direction to provide a larger image swath than that of the high-resolution (stripmap) mode. Owing to the pulse repetition frequency (PRF) timing relationship, the PRF limits the selection of the image swath. Moreover, three adjacent subswaths are continuously illuminated and combined into an image with a large swath to obtain an SAR image with a wide swath. However, the scanning ScanSAR yields a reduced synthetic aperture period than that of the stripmap mode, and the azimuth resolution is degraded. The schematic of the ScanSAR is presented on the right-hand side of Figure 2. As observed, this uses the same antenna beam as the high-resolution mode and

covers a width of approximately 5 km in all of the beams. Upon considering a 1% overlap between the adjacent beams, the wide-swath mode combines three adjacent subswaths to construct the required width of 15 km. In addition, this mode significantly reduces the power consumption by reducing the transmit duty cycle (<15%) and provides a resolution of 4 m with <160 MHz signal bandwidth.

## 3. SAR Payload Design

In relation to the SAR payload design of the S-STEP with an innovative shape, two important system aspects—parameter and architecture design processes—are described herein. Specifically, the SAR payload subsystem design process involves generating a series of system specifications. This is preceded by the mission design process that broadens the user requirements and translates to the requirements of the system element and subsystem.

The key mission parameters for the SAR user requirements of the S-STEP microsatellite include wide coverage, frequent revisits, lower Earth orbit, multimode swaths, fine resolution, small incidence angle, lightweight, small volume, and low power consumption. For the SAR system design, the fundamental performance parameters associated with the image quality are the system impulse response, spatial resolution, peak and integrated sidelobe ratio, range and azimuth ambiguity ratio, NESZ, and data rate of the SAR payload. We selected the military-grade or automotive-grade components for each unit of the SAR payload that were compatible with the space-grade parts as a low-cost approach under the NSP.

### 3.1. System Parameter Design

The SAR payload can be designed using two major approaches: performance- and constraint-driven. Because small satellites pose tight restrictions on size, weight, power, and cost (SWaP-C), the SAR payload should be designed using a constraint-driven method. To reduce the size of the SAR satellite to fit the microsatellite, the size of the SAR antenna should be reduced foremost. Alternatively, the power consumption, thermal control, and data rate should be decreased. Fundamentally, the constraints impacting the design of the SAR systems for small satellites include the size, mass, power, and cost. Moreover, the mass and size limit the area of the antenna and, consequently, limit the gain of the antenna. This, in turn, affects the sensitivity of the SAR image, as well as the azimuth resolution and swath width. Furthermore, the maximum transmission power is limited by the available DC power, and the sensitivity of the SAR image is expressed in the NESZ as follows:

$$\sigma_{NE} = \frac{4(4\pi)^3 R_s^3 L_s k T_s B_t \sin \theta_i v_s}{c \cdot G_t(\theta_e) \cdot G_r(\theta_e) \lambda^3 P_t \tau_p f_p} ,\tag{1}$$

where $L_s$ denotes the system loss, $B_t$ represents the transmitted bandwidth, $\tau_p$ denotes the pulse width, $T_s$ represents the system noise temperature, $G_t$ and $G_r$ represent the transmit and receive (Rx) antenna patterns, respectively, $P_t$ denotes the peak transmitted power, $f_p$ represents the PRF, $R_s$ denotes the slant range to the scene center, $\lambda$ denotes the wavelength, $\theta_i$ represents the incidence angle, $k$ denotes the Boltzmann constant, $c$ denotes the propagation velocity, and $v_s$ represents the satellite velocity.

As observed from this equation, the sensitivity of the SAR image can be improved by increasing the transmission output or the transmission pulse width. Alternatively, the slant range of the target can be reduced as well. Therefore, in small satellites, the incidence angle is typically set in the range of 15–35° with a small slant range owing to the limitations of the transmission power and antenna size. In another approach, the physical size of the antenna can be increased (in this case, the azimuth resolution decreases) or the antenna gain is increased. As the transmission bandwidth increases, the system complexity increases as well. Thus, the required resolution is reduced to diminish the transmission bandwidth. Moreover, the noise level can be decreased by reducing the receiver or quantization noise, which consequently minimizes the system losses. After designing the SAR system based on the stripmap mode, other modes such as the ScanSAR mode were designed based on the obtained results.

The azimuth resolution can be evaluated as follows:

$$\rho_a = \frac{\lambda}{2\theta_{int}},\tag{2}$$

where $\theta_{int}$ denotes the integration angle, which is more advantageous than the L-band and C-band for selecting the X-band and obtaining high-resolution SAR images. Because the antenna gain is inversely proportional to the square of the wavelength, the X-band selection (9.65 GHz) is extremely advantageous for small and lightweight antennas. In addition, the X-band SAR payload can facilely achieve a high resolution with a smaller satellite mass. Moreover, the X-band is suitable for recognizing the ground surface objects, and the frequency bands located higher than the Ku-band are subject to large rain attenuation. Furthermore, the technology of GaN devices such as high-power amplifiers did not mature well in higher frequency bands. Therefore, the X-band was selected for the S-STEP mission. The antenna size (antenna length) was determined using the azimuth resolution because the azimuth resolution of the stripmap mode was determined to be 1/2 of the antenna length. Therefore, the antenna length of the satellite in the S-STEP was selected as 2 m for the stripmap mode operation with a high resolution of 1 m. Based on the iterative design optimization process, the SAR payload system parameters were selected, as listed in Table 3. Based on the above description, the design workflow of the S-STEP SAR payload is illustrated in Figure 5.

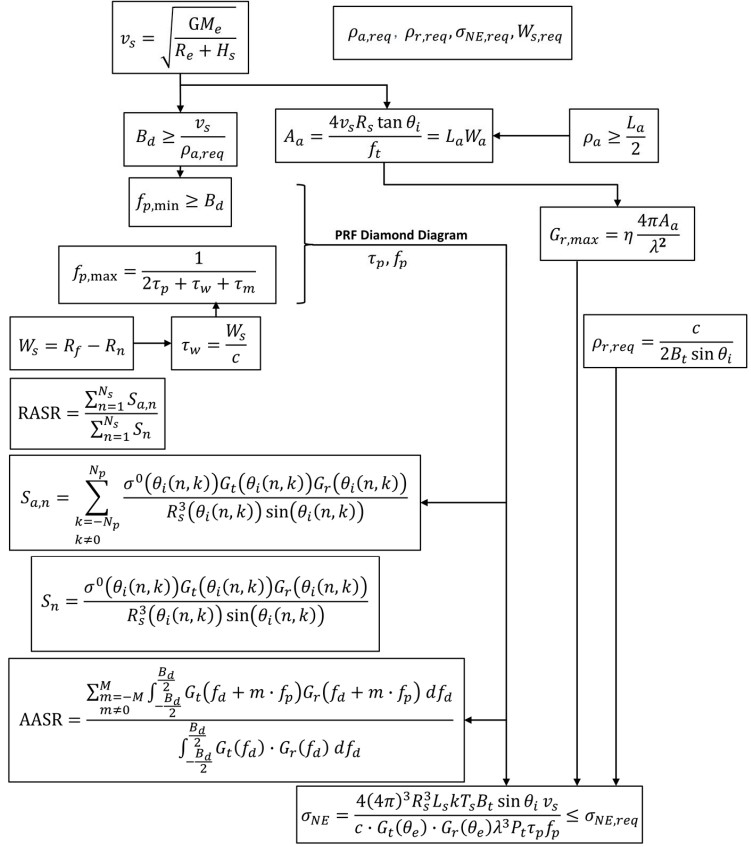

**Figure 5.** Design workflow of the SAR payload of S-STEP according to the required resolution, swath width, and NESZ sensitivity where $\rho_{r,req}$ required range resolution, $\rho_{a,req}$ required azimuth resolution, $\sigma_{NE,req}$ required NESZ, $W_{s,req}$ required swath width in range, $B_d$ Doppler bandwidth, $A_a$ antenna area, $L_a$ antenna length, $W_a$ antenna width, $f_t$ transmitted frequency, $\eta$ antenna efficiency, $f_{p,min}$ minimum PRF, $f_{p,max}$ maximum PRF, $R_n$ near range of swath, $R_f$ far range of swath, $W_s$ swath width in range, $\tau_w$ time duration of swath, $\tau_m$ guard time, $G$ universal gravitational constant, $M_e$ mass of the Earth, $R_e$ radius of the Earth, and $H_s$ satellite altitude, respectively.

**Table 3.** Key system parameters of the S-STEP SAR payload.

| Parameters | Value |
| --- | --- |
| Center frequency | 9.65 GHz |
| Polarization | single (VV) |
| Antenna size | 1970 $\times$ 1060 mm |
| Number of panels | 1 |
| Array columns (azimuth) | 4 |
| Array rows (elevation) | 48 |
| Antenna gain | 41 dBi |
| Antenna beamwidth | 0.8° in azimuth |
| | 1.6° in elevation |
| Transmitted power | 1.92 kW max |
| Average power consumption | 2.8 kW max |
| Transmitted bandwidth | 400 MHz max |
| Pulse width | 150 µs max |
| Transmit duty cycle | 30% max |
| Noise figure | 4 dB |
| Pulse repetition frequency | 2~10 kHz |
| Quantization | 10 bits |
| Date rate | 1.5 Gbps |
| System loss | 2.6 dB |
| Payload mass | 28.9 kg max |

### 3.2. System Architecture Design

The system architecture of the SAR payload was designed by applying the system parameters derived from the design process of the S-STEP SAR payload, as depicted in Figure 5. According to the original definition proposed by INCOSE, model-based system engineering (MBSE) is the formalized modeling procedure to support system requirements, design, analysis, verification, and validation activities initiating in the conceptual design phase. These should continue throughout the development and later life-cycle phases [31]. The architecture analysis and design integrated approach (ARCADIA) with its open-source model editor Capella [32,33] was selected for the present study. The adoption of digital modeling-based engineering, instead of document-oriented engineering, reduces inconsistencies and errors and facilitates the system analyses for the rapid and effective development of small satellites, such as the S-STEP, according to the NSP. The functional and logical architectural diagrams of the S-STEP SAR payload using Capella are presented in Figure 6a,b, wherein the primary functional chains of the S-STEP SAR payload are depicted in Figure 6a based on the ARCADIA approach, e.g., Tx, Rx, and frequency synthesis. The physical architecture of the S-STEP SAR payload comprising the SAR antenna unit (SAU), radio frequency unit (RFU), digital control unit (DCU), and power supply unit (PSU) is illustrated in Figure 6c.

The architectural block diagrams of the SAR system are shown in Figure 6c. The SAR system comprises a DCU for waveform signal generation and digital receiving processing; an RFU for up- and down-conversion of the RF signal; a SAU for beamforming, beam steering, high-power amplification, and radiation; and a PSU for power supply and regulation. A relatively high-transmission duty cycle of 30% can produce a higher-than-average Tx power. Based on the standpoint of the Tx amplifier, augmenting the peak power is more challenging than increasing its duty cycle under appropriate thermal management. As indicated in Table 2, the system loss of 2.6 dB comprises the transmission loss inside the satellite from the X-band TRM to the antenna input through blind-mate bullet connectors. Additionally, ohmic loss, dielectric loss, and reflection loss inside the antenna are included in the antenna gain. The system noise figure corresponds to the cascaded noise figure from the antenna input to the low-noise amplifier (LNA) via the filter, circulator, and limiter and includes the noise figure of the LNA if the antenna is facing the Earth at 290 K noise temperature [23].

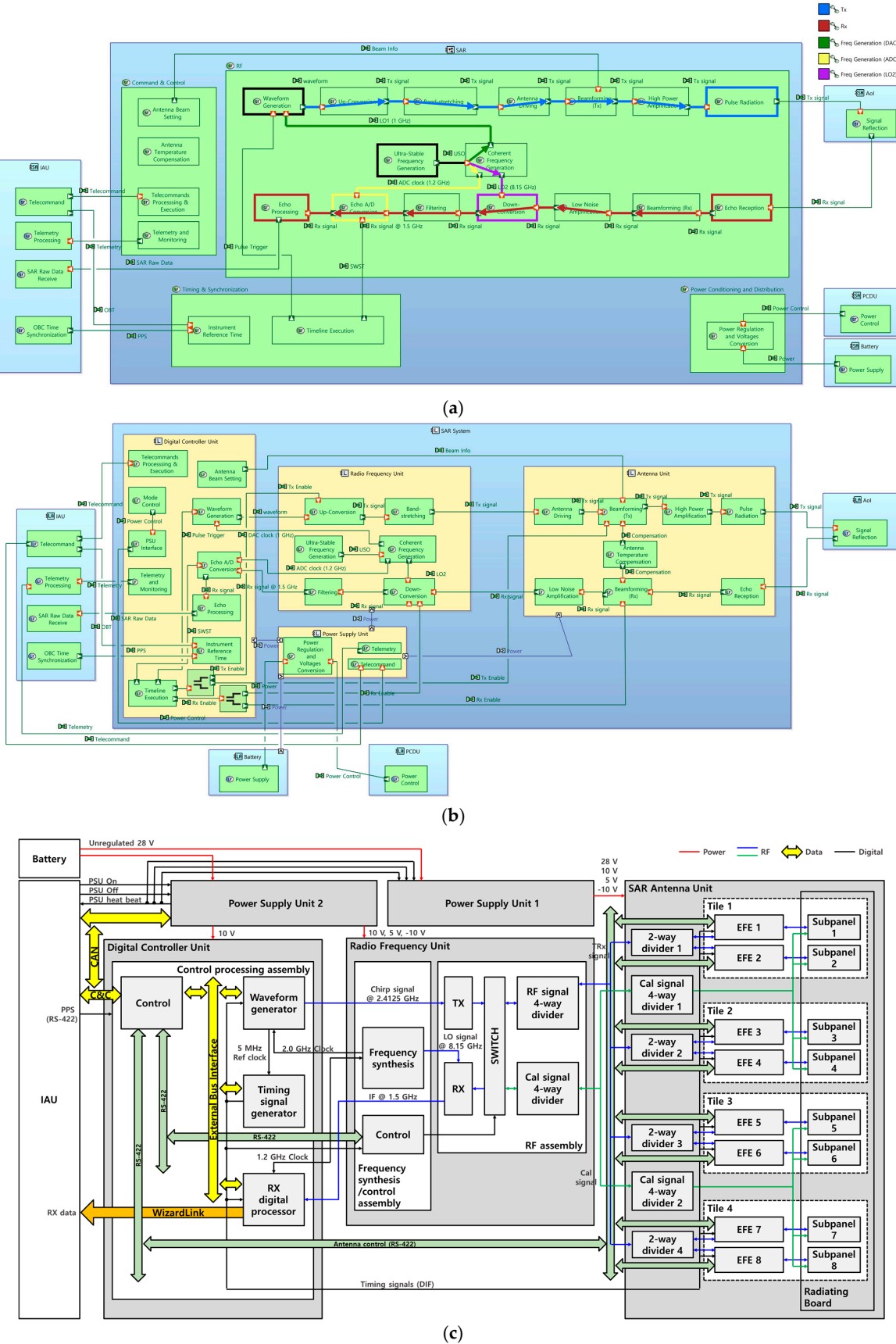

**Figure 6.** System architectural block diagrams of the S-STEP SAR payload: (**a**) functional architecture, (**b**) logical architecture, and (**c**) physical architecture.

The mode transition diagram of the S-STEP SAR payload is presented in Figure 7, which includes three operating modes—supporting, imaging, and anomaly modes—and ten operating states—off, reset, init, stand-by, stabilize states for the support mode, calibration and mission states for the imaging mode, and init anomaly, stand-by anomaly, and stabilize anomaly states for the anomaly mode. The three anomaly states correspond to the init, stand-by, and stabilized states, respectively. Moreover, an internal calibration was performed to maintain the stable performance and operating states of the SAR payload. In particular, this internal calibration mode was implemented to calibrate the amplitude and phase of the active components of the TRMs in the active phased-array antenna. Unlike external calibration, the calibration state for the internal calibration should be conducted before and after every image acquisition.

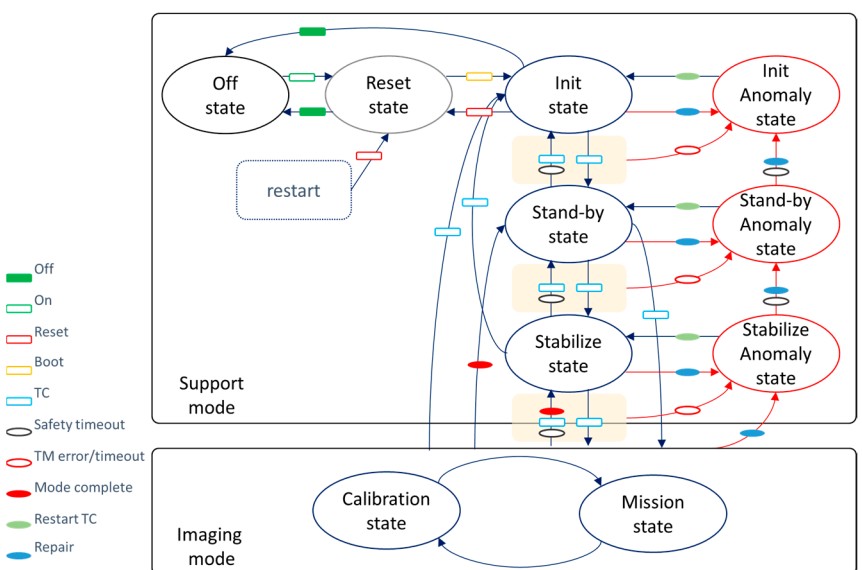

**Figure 7.** Mode transition diagram of the S-STEP SAR payload.

Based on the above-mentioned iterative design optimization process, a brief overview of the key parameters of the S-STEP SAR payload is presented in Table 2. The simulation and analysis of the SAR system design were performed using the SAR system design tool, as described in Section 6. The dimensions of the designed antenna were 1970 × 1060 mm.

The essential concepts of the innovative S-STEP satellite design include the bus–payload integrated flat-panel-type structure and highly modularized SAR antenna with a tile structure. In addition, an efficient unit design with simplified and integrated functions, a fatigue-life prediction methodology for electronic printed circuit boards (PCB), a dedicated VFOD, and a unique thermal control based on passive thermal control hardware are also applied.

The hardware and software of the SAR payload were designed to achieve the given system performance requirements. As depicted in Figure 8, the lightweight microsatellite SAR payload comprises four units: SAU, RFU, DCU, and PSU. As shown in Figure 8a, each EFE assembly of the SAU has two heat radiators for effective heat radiation of heat from each channel of TRMs as mentioned previously. In case of the lightweight S-STEP microsatellite, the data link subsystem (DLS) was integrated into one of the platform units, IAU.

The SAR antenna is one of the essential design drivers for SAR satellites, because the antenna gain, cross polarization, and sidelobes directly impact the SAR image quality, and thus influence the final imaging performance. Therefore, a lightweight, high-gain antenna with a low footprint is vital for implementing a small X-band SAR microsatellite. These requirements involve the use of deployable antennas.

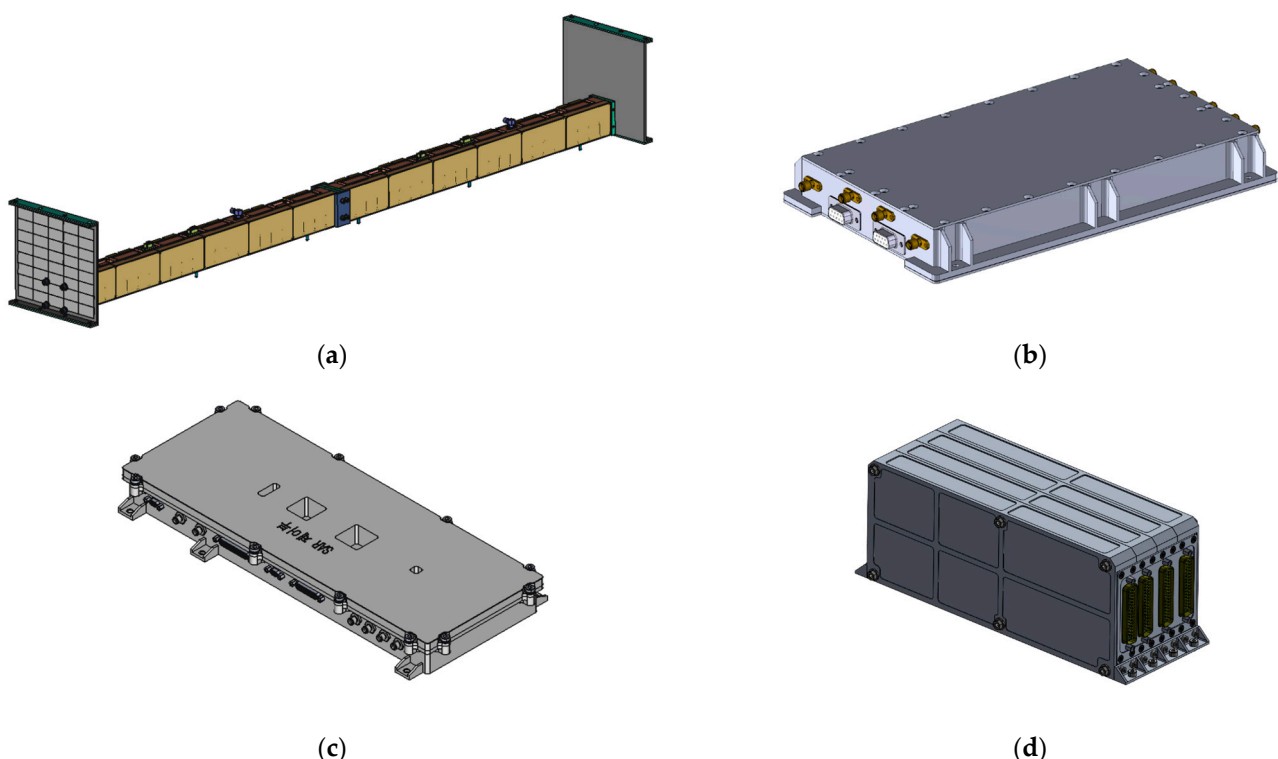

(a)

(b)

(c)

(d)

**Figure 8.** SAR payload units of S-STEP containing (**a**) integrated electronic front-end (I-EFE) assembly, (**b**) radio frequency unit, (**c**) digital control unit, and (**d**) power supply unit.

Although several promising developments are ongoing for deployable inflatable antennas, the difficulty of gas pressure control still poses serious problems for surface accuracy; therefore, this antenna is considered unsuitable for SAR microsatellites. Recently, reflect-array antennas have been mounted on the ISARA and MarCO CubeSat missions [34] that can potentially act as inexpensive, lightweight, and stowage-efficient SAR antennas but are affected by leakage issues, as demonstrated in space. Although the deployable mesh reflector antenna is ultralight and can attain high antenna gain, the deployment mechanism is extremely complicated, and the beam steering is limited [35]. Deployable active phased-array antennas are a common selection for SAR antennas, but these are generally heavy, expensive, and exhibit complex design [28]. In the S-STEP, an active phased-array antenna was applied as a single panel and integrated into the mechanical structure of the microsatellite SAR without performing the deployment mechanism to reduce the weight and maximize the stowage volume efficiency.

Overall, various SAR imaging modes such as ScanSAR can be implemented through beam steering in the microsatellite SAR using an active phased-array antenna. This enabled the operation of complex and flexible imaging scenarios. In particular, the use of an active phased-array-based antenna may be advantageous for electronic counter-countermeasures (ECCM), especially in tactical military applications [36].

However, the active antenna poses a disadvantage: the weight of the overall system increases because its development cost can be higher than that of the passive antenna, and the cooling system can become complicated owing to severe heat generation. Therefore, a precise thermal design and analysis is essential for active phased-array antennas. In context, a passive antenna that performs beam steering through maneuvering and controlling the posture of the main satellite body is occasionally used in small satellites.

Slotted waveguides and microstrip patch elements are primarily used as radiators for phased-array-type SAR antennas. The RF characteristics of the slotted waveguide-type antennas are highly suitable because distinct types of waveguides must be applied for each polarization, and the implementation of dual polarization is extremely challenging. Despite

being expensive, aluminum was selected owing to its small thermal deformation, and the carbon fiber reinforced polymers (CFRP) were applied for weight reduction. However, the microstrip patch element can be easily designed and manufactured. Additionally, it is inexpensive, lightweight, and exhibits a wider bandwidth than a slotted waveguide. In contrast, the construction of the beamforming network using a microstrip line causes relatively large losses. The losses can be minimized for performance maximization, and the antenna efficiency can be improved only through precise design/manufacturing and surface treatment. Alternatively, reflector-type antennas are deployed in a lightweight mesh or solid-type antenna.

The system architecture of the TR module can be classified into four major categories, i.e., separate transmit and receive modules, shared phase-shifter module, common-leg module, and common-leg module with calibration path architectures [37,38]. In general, the mechanical structure or configuration of the transmit/receive module can be categorized into tile (plank-type) or brick-type modules [39]. The brick-type module exhibits a planar structure with several devices mounted on a substrate, whereas the tile-type module displays a stacked structure with several stacked modules. Although the brick-type structure can be easily implemented as a flat structure, it poses the disadvantages of increased weight and size. Specifically, the tile type was implemented as an MMIC with stacked chips to enable miniaturization; however, the structure was complicated, and the development was challenging. The brick-type structure requires a wider space compared to the tile-type structure, and it was, therefore, advantageous for integrating a heat dissipation system and realizing a high transmission output. However, the tile-type structure is advantageous for compact, lightweight, scalable, and modular design owing to the development of system-on-chip technology. The S-STEP SAR payload adopted an active phased-array antenna based on a microstrip patch-array antenna, which is advantageous for miniaturization/lightweight, easy design/manufacturing, and brick-type TRM. Thus, a large space can be secured to install the SAR payload and body components on the rear of the antenna.

All electronic units were installed inside the satellite structure, specifically on the rear of the SAR antenna, as depicted in Figure 9. The design results for each unit of the SAR payload are described as follows.

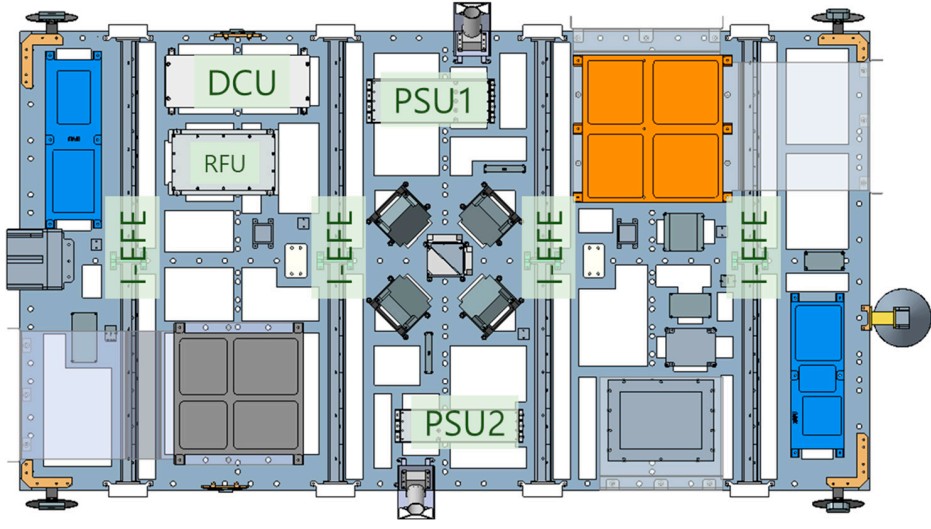

**Figure 9.** Internal layout of the S-STEP satellite units.

## 4. SAR Payload Unit Design

### 4.1. SAR Antenna Unit

The SAU was an active phased-array antenna with a mass of 21 kg and dimensions of 1970 × 1060 mm. It comprised 4 columns and 48 rows of phase centers, wherein each

active row contained a 16-element subarray. The antenna was a single polarization (VV) antenna, and the radiating subarrays were implemented using a honeycomb sandwich microstrip patch construction, which is lightweight, inexpensive, and increased the structural stiffness. In addition, the SAU comprised a radiating microstrip patch radiating board, electronic front-end (EFE), and feeding network. More specifically, the distances between the microstrip patch elements were designed as 0.965λ and 0.672λ in azimuth and elevation, respectively, considering the antenna gain, beam-steering interval, and grating lobe levels. We selected an aperture-coupled SIW cavity-backed microstrip patch-array antenna with active phased-array technology, considering the wide bandwidth with low reflection loss, inexpensiveness, lightweight, and simple fabrication for mass production.

As displayed in Figure 10, the tile-based SAR antenna structure is advantageous for shortening the development period and reducing the costs by enabling qualification tests for each antenna tile that minimizes the testing for the entire antenna unit. In addition, if the requirements of the SAR antenna (e.g., antenna gain and transmit power) have to be altered for various missions, these can be immediately reflected by simply adjusting the number of antenna tiles.

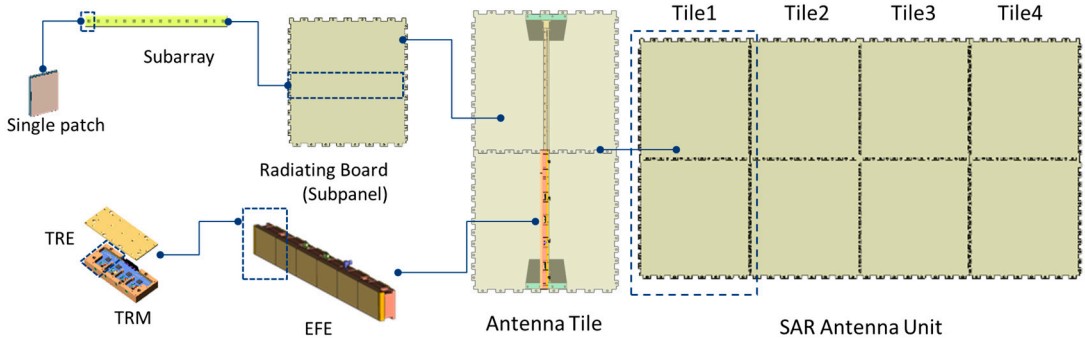

**Figure 10.** Mechanical structure of the tile-based S-STEP SAR antenna unit (SAU).

A single radiation element constituting a subarray including a test jig is presented in Figure 11a. A single radiating element with a total thickness of 4.9 mm contains an upper patch, substrate, feeding line, substrate, lower patch, substrate, and spacer, as depicted in Figure 11b. In particular, the antenna radiated electromagnetic waves through the feeding line and upper patch, and the rear radiation was reflected through the lower patch and its surrounding metal via a gain enhancement in the front direction. Additionally, the insertion of a honeycomb-based spacer between the upper patch and feeding line secured the wide bandwidth characteristics, ensured structural stability, and reduced the weight. As this type of antenna structure is a low-profile antenna based on a microstrip patch, it possesses suitable mechanical characteristics for microsatellites. Moreover, the RF-35TC thermally conductive low-loss laminate was selected for the upper and lower substrates with a dielectric constant of 3.5 and thicknesses of 0.5 and 0.508 mm, respectively. The two substrates were separated by a 3.0 mm thick quartz fiber honeycomb core, wherein the bonding sheets for the honeycomb and substrates were 3M AF163-2 and isola 185HR, respectively. According to the European Space Agency standard ECSS-Q-ST-70-02C, all these materials for the subarray are used in space (outgassing) and applied to satellites. As depicted in Figure 11c, the subarray is designed using 1 × 16 patches, considering the grating lobe and beam-steering interval. The radiating board assembly was cured under pressure in an autoclave. In particular, the antenna structure was designed and optimized using the commercial electromagnetic (EM) simulation software CST Microwave Studio. The antenna was designed using a honeycomb-structured spacer for weight reduction and structural stiffness.

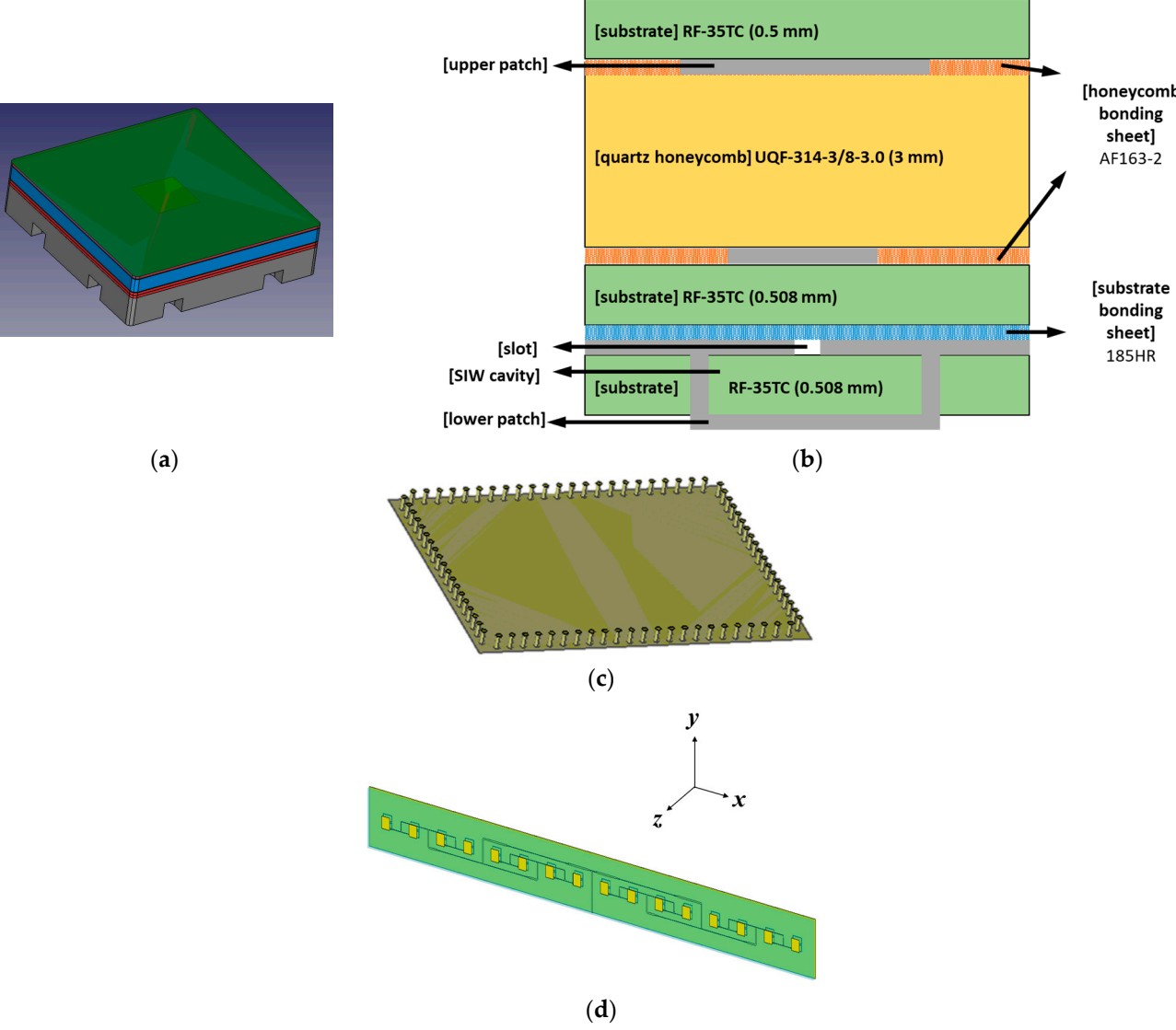

**Figure 11.** (**a**) Configuration of an element antenna, (**b**) sectional structure, and (**c**) SIW cavity of the radiating board, and (**d**) configuration of a 1 × 16 subarray of the S-STEP SAR antenna.

Although the existing TRM for satellites is developed using GaAs-based high-power amplifiers (HPAs), recent studies have applied GaN-based HPAs to TRM for satellites. In comparison to GaAs HPAs, GaN-based HPAs can operate at a higher supply voltage and exhibit a high-power density. Therefore, a high-power amplification can be achieved with a small area. Additionally, it has a low thermal resistance and exhibits appropriate thermal properties.

The external configuration and an exploded view of the EFE are presented in Figure 12a,b, respectively. As depicted in Figure 12c, the block diagram of the single-channel TRM using a 15 W X-band GaN HPA MMIC (UMS CHA8610-99F) is fabricated using the pseudomorphic high electron mobility transistor (pHEMT) process [40]. Specifically, the EFE contained six 4-channel TRMs and a control assembly. The transmission path of the TRM comprised an amplifier to drive the HPA and a GaN HPA for the high-power amplification of the transmission signal. In contrast, the receiver path contained a limiter (TriQuint TGL2201) to prevent damage to the low-noise amplifier (LNA) and a GaAs LNA MMIC (UMS CHA2110-98F) for low-noise amplification of the received signal. Additionally, the common path included a core-chip component to control the amplitude and phase of the transmit/receive signal and a circulator component connecting to the

antenna. Moreover, the core chip controlled the amplitude and phase of the Tx and Rx signals. The 6-bit X-band core-chip GaAs MMIC (Ommic CGY2170YHV/C1) is a common MMIC containing a 6-bit attenuator, 6-bit phase shifter, and switches.

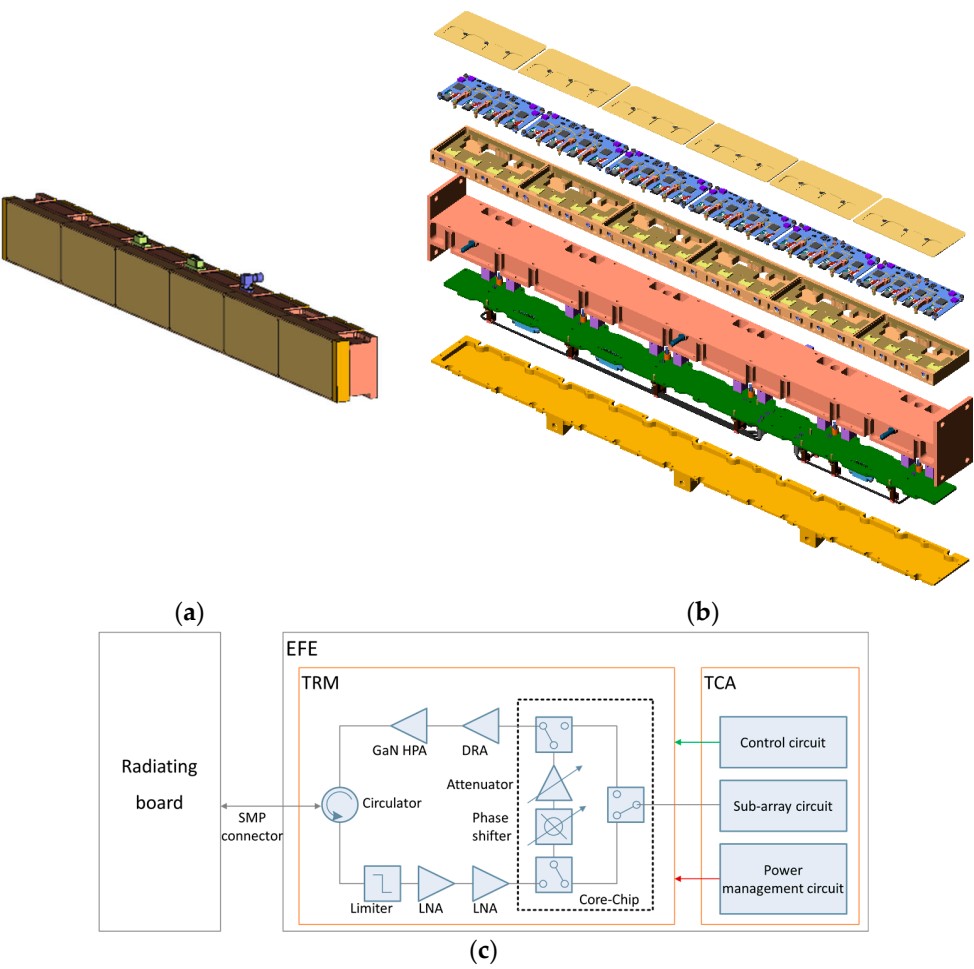

**Figure 12.** (**a**) Configuration of the electronic front-end (EFE) assembly, (**b**) exploded view of the EFE, and (**c**) architecture of the transmit/receive module (TRM) of the S-STEP SAR antenna unit.

In recent T/R module research studies, the scholars used multilayer substrates composed of low-temperature co-fired ceramic (LTCC). In addition, hermetic sealing is used in aerospace applications to protect T/R modules against environmental factors. These aspects ensure the compactness and robustness of the T/R module; however, they require a complex and costly fabrication process. In the S-STEP SAR payload, the TRM was designed with a low-cost heterojunction (ceramic and FR4) multilayer substrate instead of an LTCC substrate. Furthermore, the TRM was sealed using a laser welding process. Eventually, the designed heterojunction multilayer substrate for the TRM comprised three FR4 substrates (185HR) for digital signals and two ceramic substrates (RO6035HTC and RO4450F) for RF signals.

The control assembly separately configured from the TRM performed power supply voltage conversion, power supply control, amplitude and phase control for beam steering, and distribution and combination of transmit/receive signals, respectively. In particular, a single TRM contained four channels for size and weight reduction and was manufactured using materials applicable in space, such as for outgassing and satellites, adhering to ECSS-Q-ST-70-02C.

The T/R module housing material was AL6061, which is lightweight, inexpensive, and exhibits suitable heat conduction characteristics. However, this material has a different

thermal expansion coefficient as compared to the GaAs HPA MMIC. Accordingly, GaAs MMIC chips were attached using H20E epoxy at 120 °C. The thermal analysis results of the SAR antenna, including the TRM, were detailed in a prior study on the S-STEP [26].

### 4.2. Radio Frequency Unit

As displayed in Figure 13a, the implemented RFU had a mass of 1.5 kg, DC power consumption of 30 W, and comprised transmit, receive, frequency synthesis, control, and power circuits. Military-grade components compatible with the space-grade parts were selected for each circuit. The overall functional block diagram of the RFU is illustrated in Figure 13b. The transmission signal was amplified by a factor of four in the transmission circuit and up-converted from the S-band (2.4125 GHz) to the X-band (9.65 GHz) for band stretching of the Tx waveform, and the reception signal was converted to the L-band (1.5 GHz) using the mixer in the reception circuit. The frequency synthesis circuit generated an L-band (1.2 GHz analog-to-digital converter (ADC) clock), S-band (2 GHz digital-to-analog converter (DAC) clock), and X-band (LO signal) signals using a space-grade ultrastable 10 MHz oven-controlled crystal oscillator (OCXO) (Axtal AXIOM75SL). Additionally, the control circuit controlled the transmission/reception status and signal gain using a microcontroller unit (MCU) (ATMEL ATmega128L-8A).

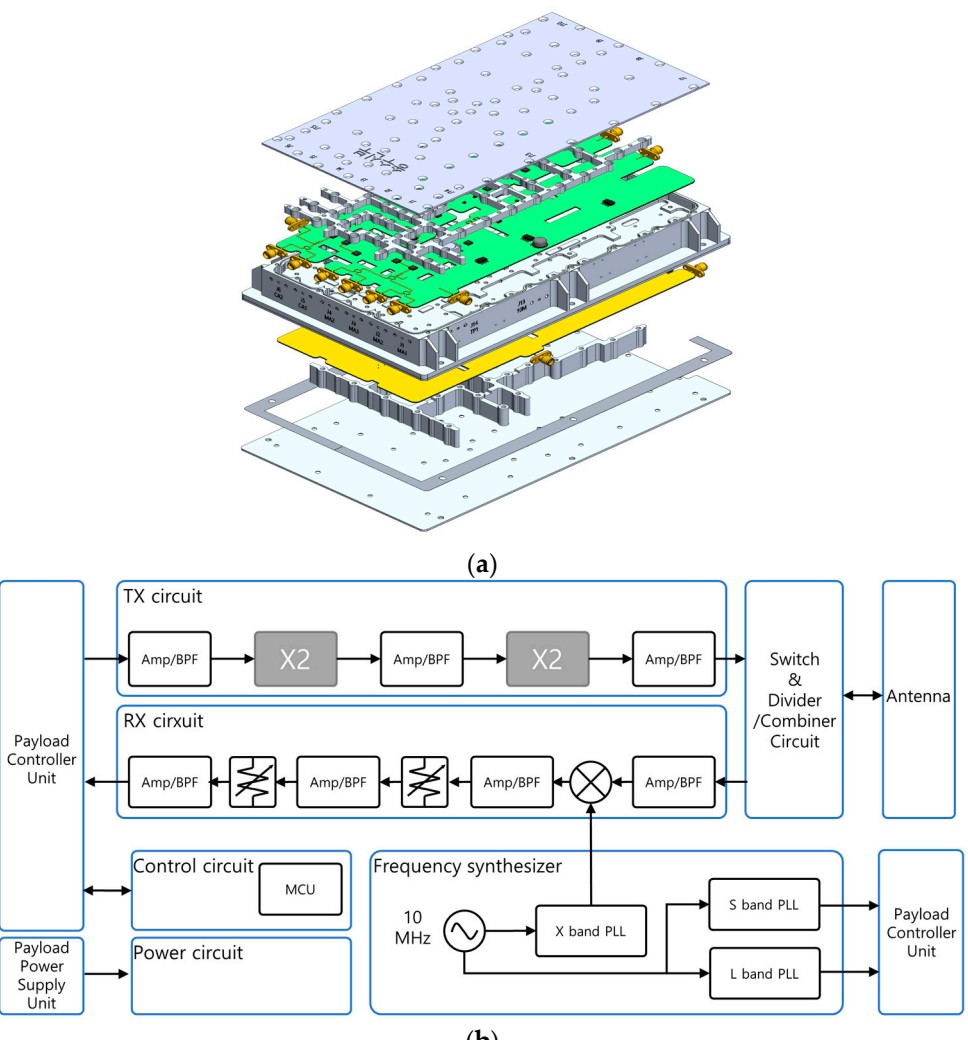

(**a**)

(**b**)

**Figure 13.** (**a**) Exploded view and (**b**) architectural block diagram of the S-STEP RFU.

In particular, the five circuits in the transceiver were integrated into a single module to reduce the weight to suitable limits for a miniature SAR satellite. An expanded view of the integrated module of the S-STEP RFU is presented in Figure 13a. A PCB with integrated transmit and receive functions and a PCB with frequency synthesis and control functions were fastened to the upper and lower surfaces of the double-sided machined housing, respectively. Moreover, bulkheads were used for the internal electromagnetic shielding.

Heat dissipation and material selection are crucial for withstanding the space environment. Thus, a thermal pad was applied between the housing and bottom to dissipate heat to the bottom, and a PCB base satisfying the outgassing standard was selected. Additionally, the inner surface of the housing was treated with chromate and the outer surface was treated with black anodizing to prevent corrosion.

### 4.3. Digital Control Unit

As depicted in Figure 14, the implemented DCU had a mass of 0.9 kg and DC power consumption of 35 W. The external and internal configurations of the DCU are presented in Figure 14a,b, respectively. The DCU of the micro-SAR satellite largely controlled the payload components, generated a wideband transmission waveform following the PDDS method with a 12-bit DAC chip (Teledyne EV12DS130BMGC), compensated the waveform distortion using a pre-distortion method, and converted and processed digital reception signals using a 12-bit analog-to-digital converter chip (Texas Instruments ADC12DJ3200). In addition, the DCU determined the SAR payload operation modes according to the command received from the ground object through the platform (Figure 8); controlled the SAU (EFE), RFU, and PSU; and acquired the status information of each SAR payload unit and reported it to the platform. Subsequently, a wideband LFM signal was generated for image acquisition, and the received raw data were formatted as Consultative Committee for Space Data Systems (CCSDS) using high-speed analog-to-digital conversion of the received signal and were transmitted to the platform.

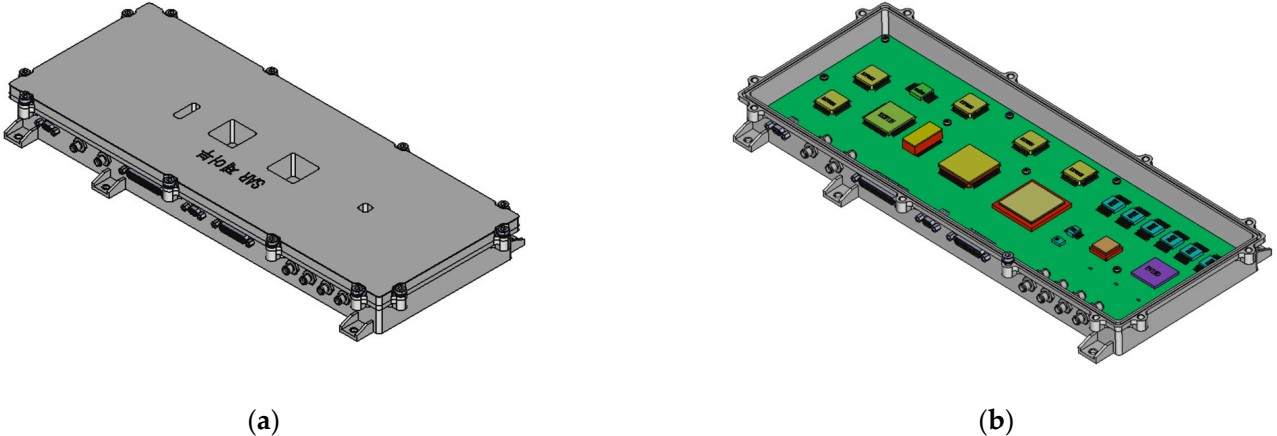

(**a**)  (**b**)

**Figure 14.** (**a**) External configuration and (**b**) internal configuration of the S-STEP DCU.

The DCU hardware architecture primarily contained a high-performance flash-memory -based FPGA and an MCU, as shown in Figure 15, which are the microchip radiation-tolerant PolarFire FPGA RTPF500TS and Microchip Radiation-Tolerant 32-bit ARM Cortex-M7 MCU SAMV71Q21RT, respectively.

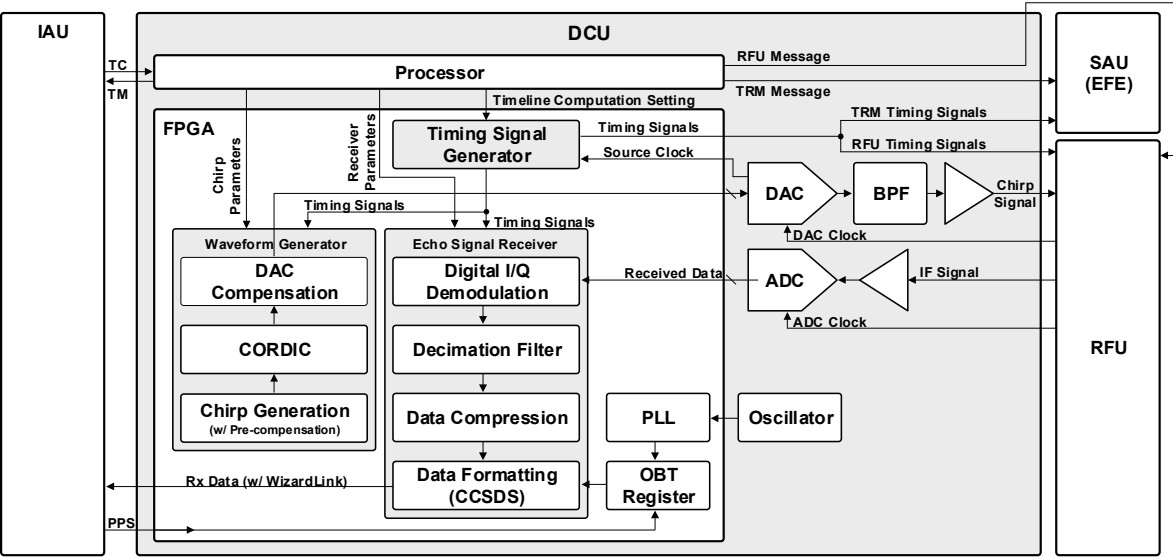

**Figure 15.** Functional block diagram of the S-STEP DCU.

### 4.4. Power Supply Unit

As portrayed in Figure 16, the implemented PSU has a mass of 5.5 kg and a DC power consumption of 300 W. The external configuration and an exploded view of the PSU are displayed in Figure 16a,b, respectively. The SAR payload separately configured the PSU to supply stable power to the DCU, RFU, and EFE units. The PSU received unregulated +28 V power from the battery and provided +28, ±10, and +5 V power to the SAR payload units.

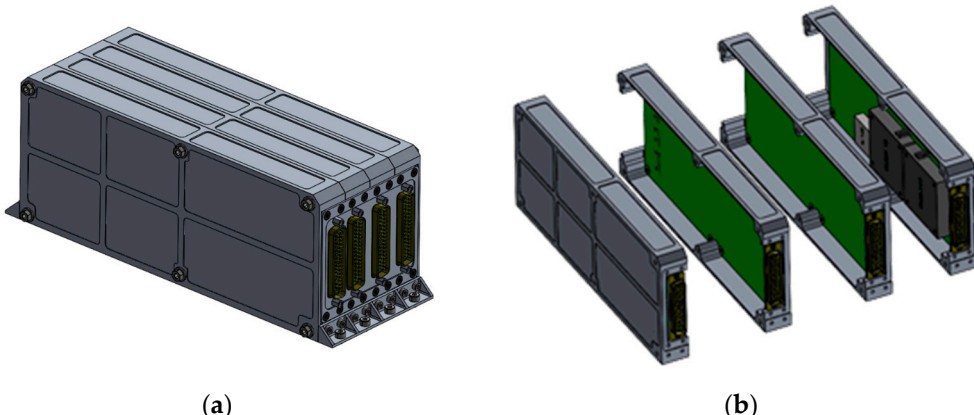

| (**a**) | (**b**) |
|:---:|:---:|

**Figure 16.** (**a**) External configuration and (**b**) exploded view of the S-STEP PSU.

### 5. Antenna Model

The necessity of an antenna model is justified for three major reasons [41–44]. First, as the number of antenna beams required for the SAR operation increases, an antenna model is needed to effectively shorten the calibration time of the antenna beam. In addition, a precise antenna model is required to satisfy and maintain the radiometric performance of the SAR images. Finally, the antenna model is very helpful in reducing the time of the in-orbit commissioning phase of the satellite and facilitating the calibration/validation (CAL/VAL) activity that occurs regularly.

In the case of the S-STEP SAR satellite, hundreds of antenna beams (>40 beams in high-resolution mode) are required to acquire high-quality SAR images in various modes. An antenna model capable of appropriately modeling these beams can accurately determine the numerous antenna patterns required for image acquisition. In most satellite SARs based on active phased-array antennas, such as TerraSAR-X, CosmoSky-Med, and Sentinel-1,

antenna models have been developed and applied. A satellite SAR based on an active phased-array antenna can flexibly form beams through excellent beam agility.

According to the results of the on-ground characterization process, the antenna characteristics can be accurately modeled by applying mathematical models. The antenna—1970 mm in length and 1060 mm in width—contains 192 microstrip patch subarrays arranged in 4 tiles in the azimuth direction (columns), each with 48 subarrays (rows), as depicted in Figure 17, wherein the blue and green squares represent patches and subarray ports connected to the TRMs, respectively. In particular, the nominal antenna pointing was 23.7° from the nadir. Moreover, the right- and left-looking acquisitions were realized using satellite roll maneuvers. The individual subarrays were connected to a single channel of the TRM, and its amplitude and phase can be adjusted by applying complex excitation coefficients. This enabled the beam steering and formation in the azimuth and elevation directions.

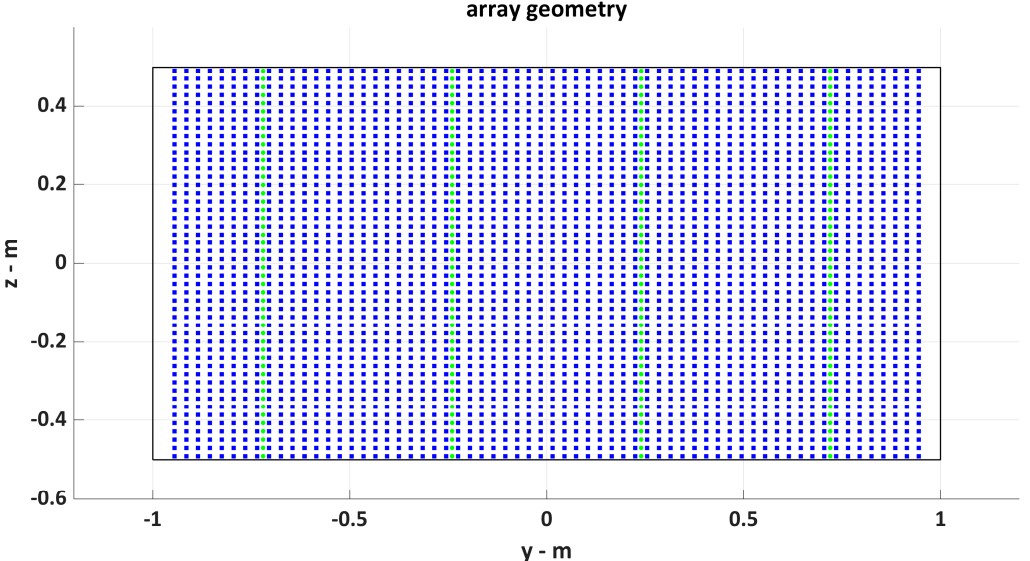

**Figure 17.** Array geometry of the S-STEP SAR antenna model.

The antenna model is primarily used to generate the antenna beam pattern data required for compensating the characteristics of each antenna beam. Specifically, the elevation beam pattern is obtained by measuring the gain variations in the distance direction with a signal measured for an area such as the Amazon Forest. Furthermore, the azimuth beam pattern is acquired by accurately estimating the Doppler spectrum as the azimuth beam pattern appeared therein. In addition, the antenna model can optimize the beam excitation coefficients to enhance the SAR image quality, such as the NESZ and distributed target-to-ambiguity ratio, for full performance at the required level.

The antenna model determines the final antenna beam pattern by multiplying the beam excitation coefficients with the weight of the beam pattern of each subarray using geometric information related to the antenna. The drift errors or failure factors occurring in each antenna element can be included in the antenna model as an error matrix.

Before testing, the antenna patterns are simulated using a detailed antenna electric model developed based on component test data. Subsequently, the antenna model is used for beam optimization to enhance the SAR performance. During the antenna pattern test phase, the model is verified and amended by comparing the tested and simulated patterns.

The antenna model mathematically evaluates the radiation patterns based on the superposition of the four inputs: subarray radiation patterns, beam excitation coefficients of each TRM, exact array geometry, and the actual state of the SAR antenna, including the

drifting or failure of the TRMs. The following equation for a phased array provides the model for the antenna radiation pattern $G_a(\theta, \phi)$ [41,42]:

$$
\begin{aligned}
G_a(\theta, \phi) = \sum_{m=1}^{M} \sum_{n=1}^{N} A_{m,n} \cdot G_{a,sub}(\theta, \phi) \cdot E_{m,n} \\
\cdot \exp\left[ jk \sin\theta \cos\phi \left( \frac{N+1}{2} - n \right) \Delta x \right] \\
\cdot \exp\left[ jk \cos\theta \sin\phi \left( \frac{M+1}{2} - m \right) \Delta y \right]
\end{aligned}
\tag{3}
$$

where $A_{m,n}$ denotes the beam excitation coefficient for the $m$-th row and $n$-th column TRM, $G_{a,sub}(\theta, \phi)$ represents the subarray pattern, $E_{m,n}$ denotes the error matrix for the $m$-th row and $n$-th column TRM, $\Delta x$ and $\Delta y$ represent the distances between the subarrays in the $x$- and $y$-directions, respectively, and $k = \frac{2\pi}{\lambda}$ denotes the wavenumber. Figure 18a,b show the element antenna patterns in azimuth and elevation, respectively. The subarray pattern $G_{a,sub}(\theta, \phi)$ can be calculated as follows:

$$
G_{a,sub}(\theta, \phi) = G_{a,az}(\theta) \cdot G_{a,el}(\phi),
\tag{4}
$$

where $G_{a,az}$ and $G_{a,el}$ denote the antenna beams in azimuth and elevation, as depicted in Figure 19a,b, respectively. Figure 20a,b show the total antenna patterns in azimuth and elevation, respectively.

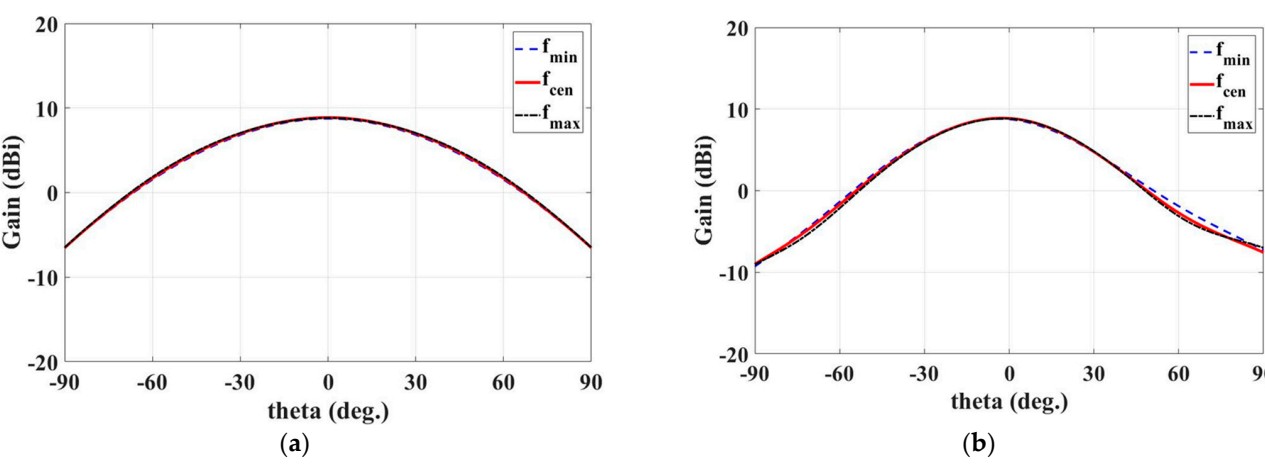

**Figure 18.** Element antenna patterns of S-STEP SAR payload in (**a**) azimuth and (**b**) elevation.

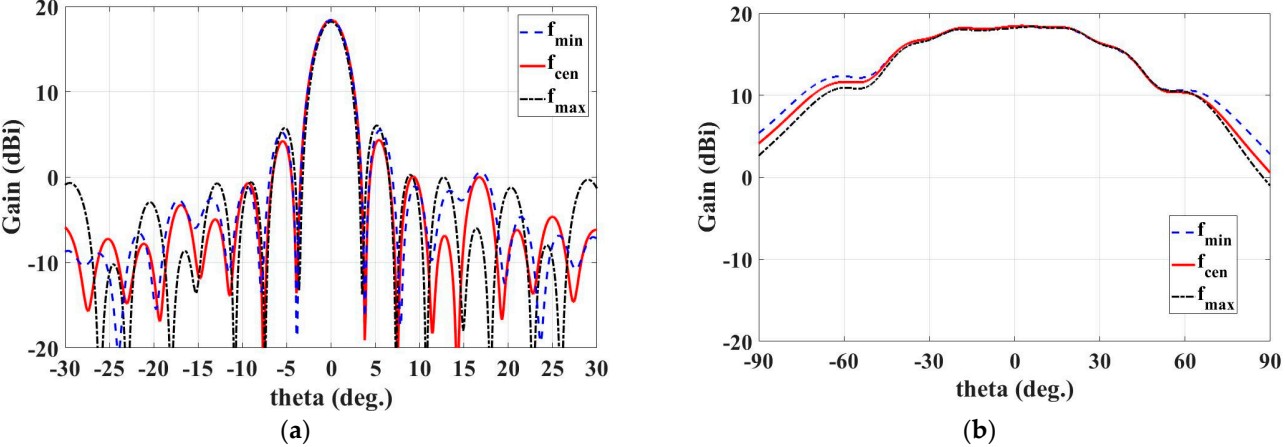

**Figure 19.** Subarray antenna patterns of the S-STEP SAR payload in (**a**) azimuth and (**b**) elevation.

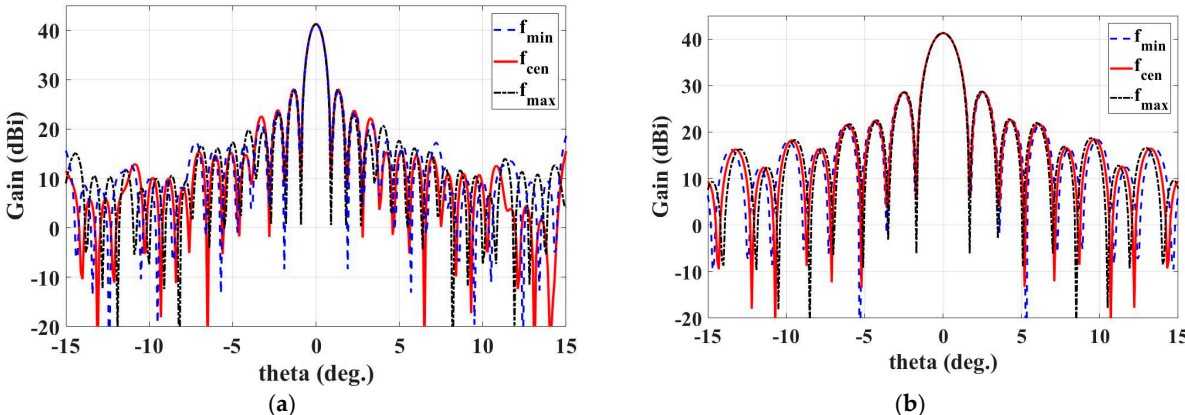

**Figure 20.** Antenna patterns of the S-STEP SAR payload in (**a**) azimuth and (**b**) elevation.

## 6. SAR Payload Performance Analysis

The predicted performance of the S-STEP SAR sensor was determined via simulation during the system design phase. The performance of the SAR payload was accurately calculated prior to the satellite launch based on its ground test results and was subsequently verified by in-orbit tests and evaluations. Herein, the imaging mode design performance is presented in terms of the key performance parameters such as the antenna size, swath width selection, system impulse response function, peak and integrated sidelobe level, azimuth and range ambiguity ratio, NESZ, and data rate.

### 6.1. SAR Payload System Design Tool

A system design tool was developed for the S-STEP SAR payload to aid the system design and evaluate its imaging performance based on a mathematical model [2–4]. The input parameters included antenna gain, transmitted power, noise figure, etc. In particular, the SAR payload system design tool could analyze the PRF timing diagram, NESZ, and the impulse response function for various SAR modes.

The graphical user interface of the SAR payload system design tool for S-STEP is displayed in Figure 21 which depicts the system parameter settings, analysis plots (one major and two subplots), and the SAR payload operational parameters table. In addition, the collection geometry and access region of the S-STEP SAR satellite for performance evaluation are depicted in Figure 22a,b, respectively, where $\eta$ denotes the incidence angle, $h$ represents the satellite altitude, and $\alpha$ indicates the off-nadir angle. The key performance parameters of the S-STEP SAR payload for the high-resolution and wide-swath modes are detailed in Table 3.

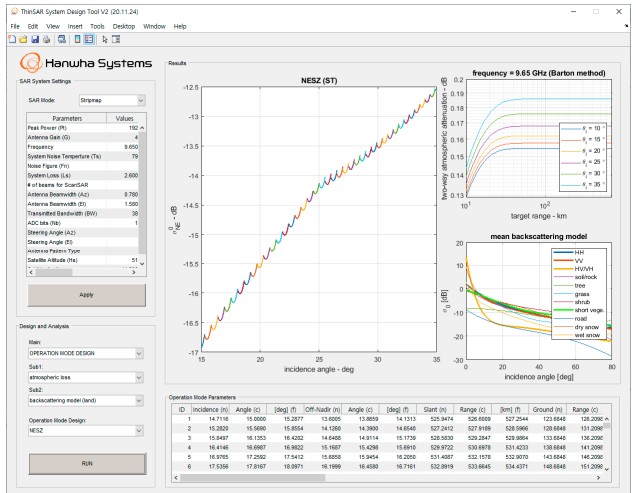

**Figure 21.** System design tool of the S-STEP SAR payload.

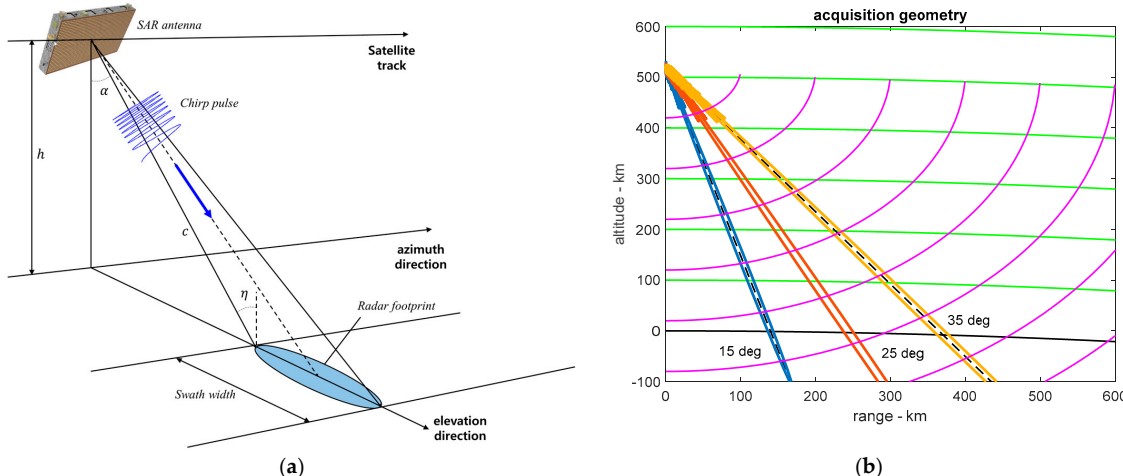

**Figure 22.** (**a**) Collection geometry and (**b**) access region of the S-STEP SAR payload.

### 6.2. PRF Diamond Diagram

An appropriate PRF range of the SAR should be selected for a given orbit using the timing diagram—the PRF diamond diagram, in the case of a spaceborne SAR considering several aspects such as the nadir echo. Accordingly, the PRF of the SAR payload was selected such that it was larger than the Doppler bandwidth, and the pulse repetition interval (PRI; reciprocal of the PRF) was larger than the reception window length corresponding to the swath width. Moreover, the PRF was selected to avoid Tx eclipsing and off-nadir returns. The diamond diagrams of the S-STEP SAR payload in the high-resolution and wide-swath modes are illustrated in Figure 23a,b, respectively, wherein the black-solid line indicates the appropriate PRF range based on the incidence angle. Additionally, the green and blue zones indicate the off-nadir returns and Tx transmission eclipsing, respectively, which should be avoided. Furthermore, the swath width was selected using the PRF diamond diagram to produce an initial set of swaths covering the access region. We assumed a guard time of 0.2 µs and a maximum transmit duty cycle of 30% for evaluating the off-nadir and transmit (Tx) pulse eclipsing regions. The antenna electronically steered the beam such that it pointed toward swaths between incidence angles of 15 and 35°. Comprehensively, the designed operational parameters of the S-STEP SAR payload in the high-resolution and wide-swath modes are presented in Tables 4 and 5, respectively.

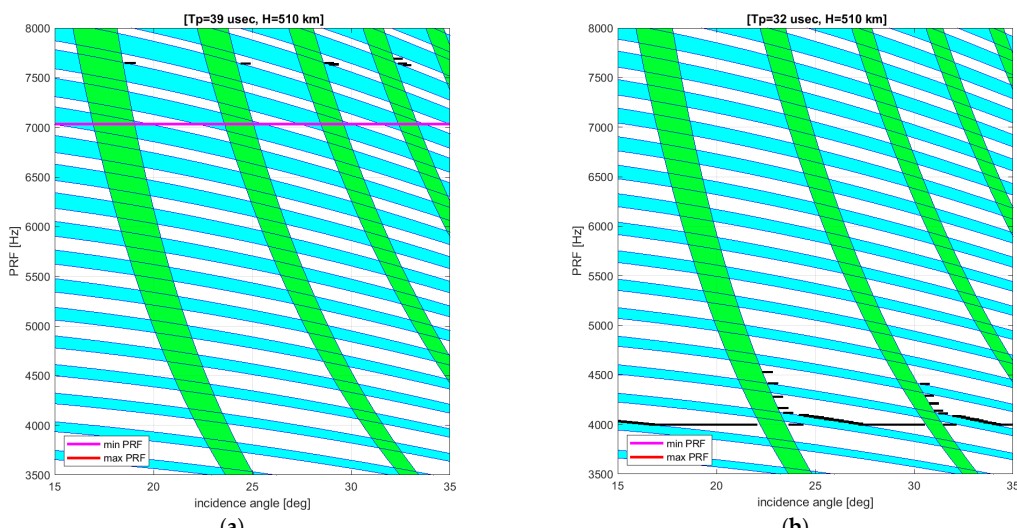

**Figure 23.** PRF diamond diagrams of the S-STEP SAR payload in the (**a**) high-resolution and (**b**) wide-swath modes.

**Table 4.** Operational parameters of the S-STEP SAR payload in the high-resolution mode.

| Parameters | Incidence Angle | | |
| | 15° | 25° | 35° |
| --- | --- | --- | --- |
| Beam steering angle | −8.02° | 1.41° | 10.29° |
| Pulse width | 41 μs | 41 μs | 41 μs |
| Transmitted bandwidth | 380 MHz | 376.27 MHz | 278.22 MHz |
| Sampling frequency | 456 MHz | 451.52 MHz | 333.86 MHz |
| PRF | 7220.39 Hz | 7105.39 Hz | 7197.39 Hz |
| Sampling window length | 49.72 μs | 55.38 μs | 60.37 μs |
| Rank | 25 | 26 | 29 |
| Sampling window start time | 46.33 μs | 65.64 μs | 46.39 μs |
| Synthetic aperture time | 0.99 s | 1.05 s | 1.15 s |
| Number of pulses acquired | 7116 | 7439 | 8249 |

**Table 5.** Operational parameters of the S-STEP SAR payload in the wide-swath mode.

| Parameters | Incidence Angle | | |
| | 15° | 25° | 35° |
| --- | --- | --- | --- |
| Beam steering angle | −8.02° | 1.41° | 10.29° |
| Pulse width | 32 μs | 32 μs | 32 μs |
| Transmitted bandwidth | 156.65 MHz | 94.07 MHz | 69.55 MHz |
| Sampling frequency | 187.98 MHz | 112.88 MHz | 83.47 MHz |
| PRF | 4033 Hz | 4068 Hz | 4000 Hz |
| Sampling window length | 40.72 μs | 46.38 μs | 51.37 μs |
| Rank | 14 | 15 | 16 |
| Sampling window start time | 37.38 μs | 37.52 μs | 75.63 μs |
| Synthetic aperture time | 0.25 s | 0.26 s | 0.29 s |
| Number of pulses acquired | 994 | 1065 | 1146 |

*6.3. Noise Equivalent Sigma Zero*

The NESZ is an index of the radiometric sensitivity of the system and reflects a quantity directly related to the SAR imaging performance, which is defined as the target radar cross-section if the signal-to-noise ratio of the image is equal to one [2–4]. The NESZ of the *n*-th antenna beam corresponding to the *n*-th swath can be expressed as [35,45]

$$\sigma_{NE,n} = \frac{4(4\pi)^3 R_{s,n}^3 L_s k T_s B_{t,n} \sin\theta_{i,n} v_s}{c \cdot G_{t,n}(\theta_e) \cdot G_{r,n}(\theta_e) \lambda^3 P_t \tau_{p,n} f_{p,n}}, \tag{5}$$

where $B_{t,n}$ represents the transmitted bandwidth, $\tau_{p,n}$ indicates the pulse width, $G_{t,n}$ and $G_{r,n}$ represent the Tx and Rx antenna pattern of the *n*-th antenna beam, $f_{p,n}$ reflects the PRF, $R_{s,n}$ denotes the slant range to the scene center of the *n*-th antenna beam, $\theta_{i,n}$ denotes the incidence angle of the *n*-th antenna beam. Based on the system parameters listed in Tables 3–5, the NESZ for the S-STEP SAR in the high-resolution and wide-swath modes are evaluated over a range of incidence angles, as displayed in Figure 24a,b, respectively.

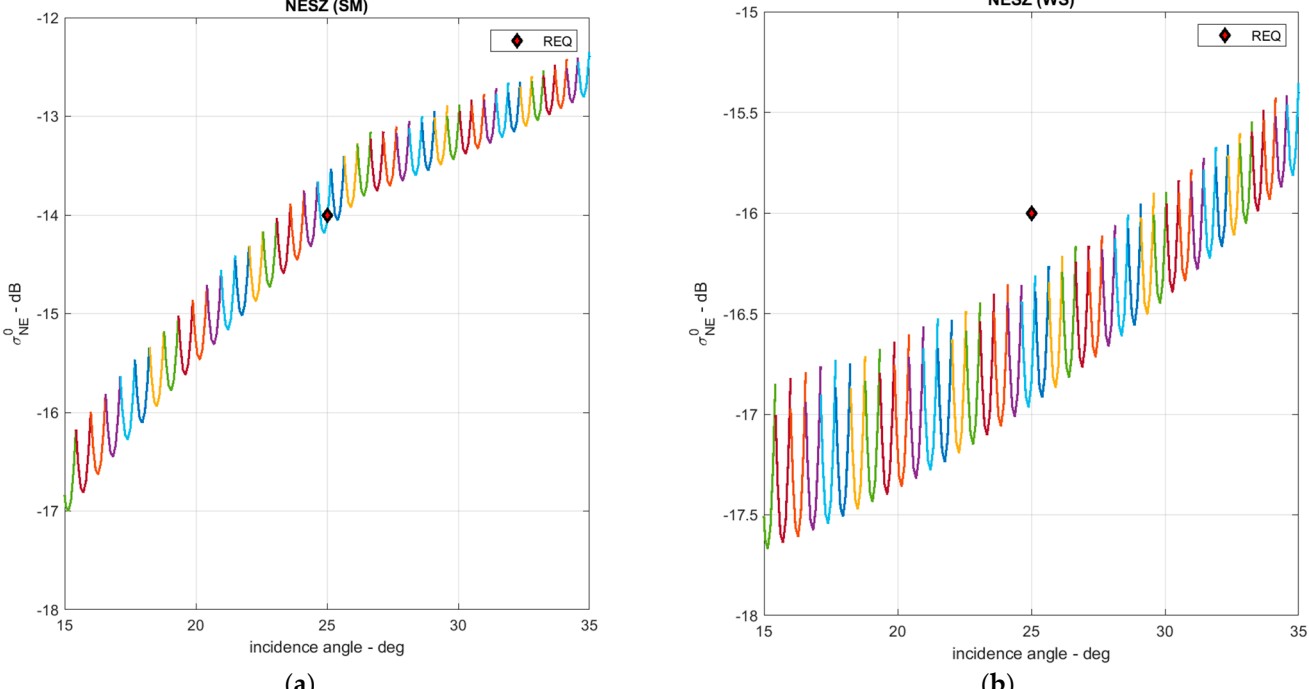

**Figure 24.** NESZ of the S-STEP SAR payload in the (**a**) high-resolution and (**b**) wide-swath modes. NESZ is lower than the required level of −14 and −16 dB at an incidence angle of 25° for the high-resolution and wide-swath modes, respectively. Each color shows the corresponding swath.

### 6.4. Impulse Response Function

The SAR image performances are characterized by an impulse response function to a point target, in terms of the range and azimuth resolutions, peak sidelobe ratio (PSLR) and integrated sidelobe ratio (ISLR). The single-look impulse response function can be expressed as

$$h_r(y) = \| \frac{1}{\tau_p} \int_{t_1}^{t_2} F_r\big(t_f(y), t_s\big) \, dt_s \|^2, \tag{6}$$

where the integrand function in range $y$ can be written as

$$F_r(t_f, t_s) = w_r(t_f) \cdot A(t_s) \exp\left\{ j\left[ \theta(t_f) + \theta(t_s) + \pi \frac{B_{t,n}}{\tau_{p,n}} t_f t_s \right] \right\}, \tag{7}$$

where $w_r(t_f)$ is a window function in range, $t_f$ is the fast time, $t_s$ is the slow time, and, in azimuth $x$ as

$$h_a(x) = \| \frac{1}{T_{sa}} \int_{-\frac{T_{sa}}{2}}^{\frac{T_{sa}}{2}} F_a\big(t_f, t_s(x)\big) \, dt_f \|^2, \tag{8}$$

where $T_{sa}$ is the synthetic aperture time, the integrand function in azimuth is

$$F_a(t_f, t_s) = w_a(t_s) \cdot A(t_f) \exp\left\{ j\left[ \theta(t_f) + \pi \frac{B_d}{T_{sa}} t_f t_s \right] \right\}, \tag{9}$$

where $w_a(t_s)$ is a window function in azimuth.

Furthermore, the expected performances of the SAR system were confirmed to satisfy the requirements in both the high-resolution (stripmap) and wide-swath (ScanSAR) modes with 5% HW error, as depicted in Figure 25a,b for the high-resolution mode and in Figure 26a,b for the wide-swath mode.

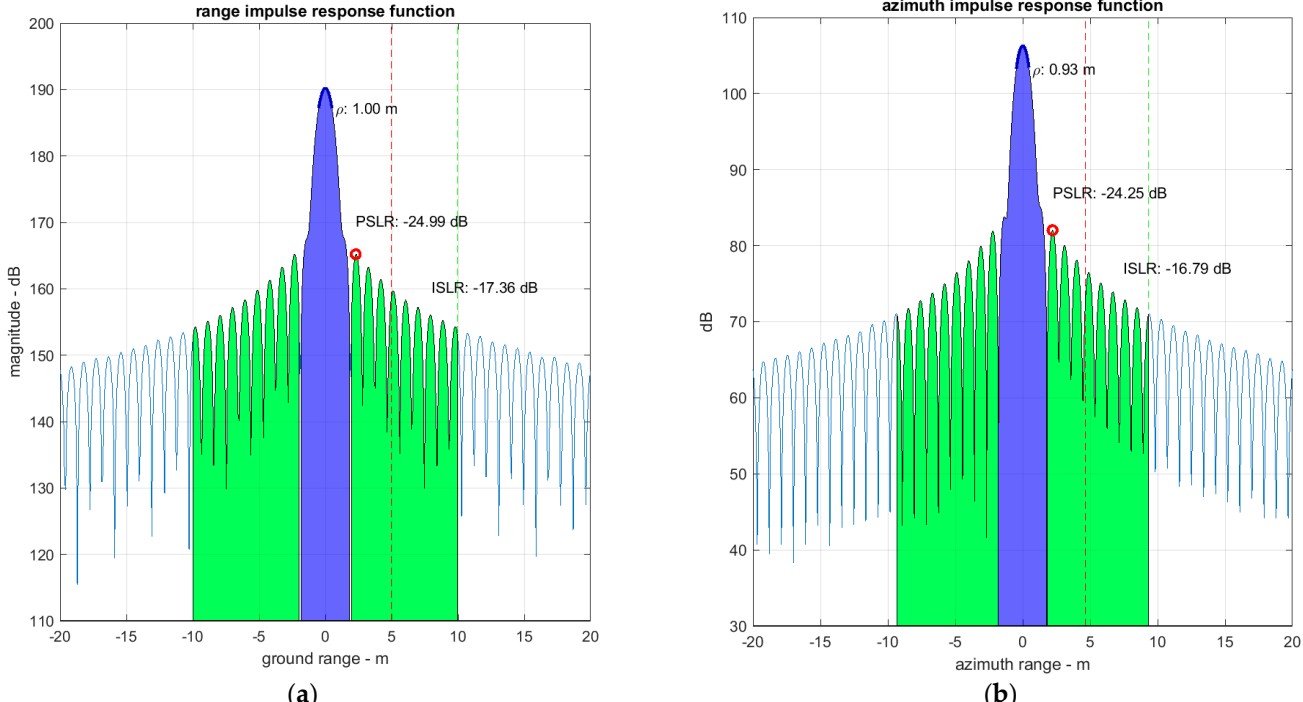

**Figure 25.** Impulse response function analysis in the (**a**) range and (**b**) azimuth of the S-STEP SAR payload in the high-resolution mode. The red circles indicate the peak side lobes.

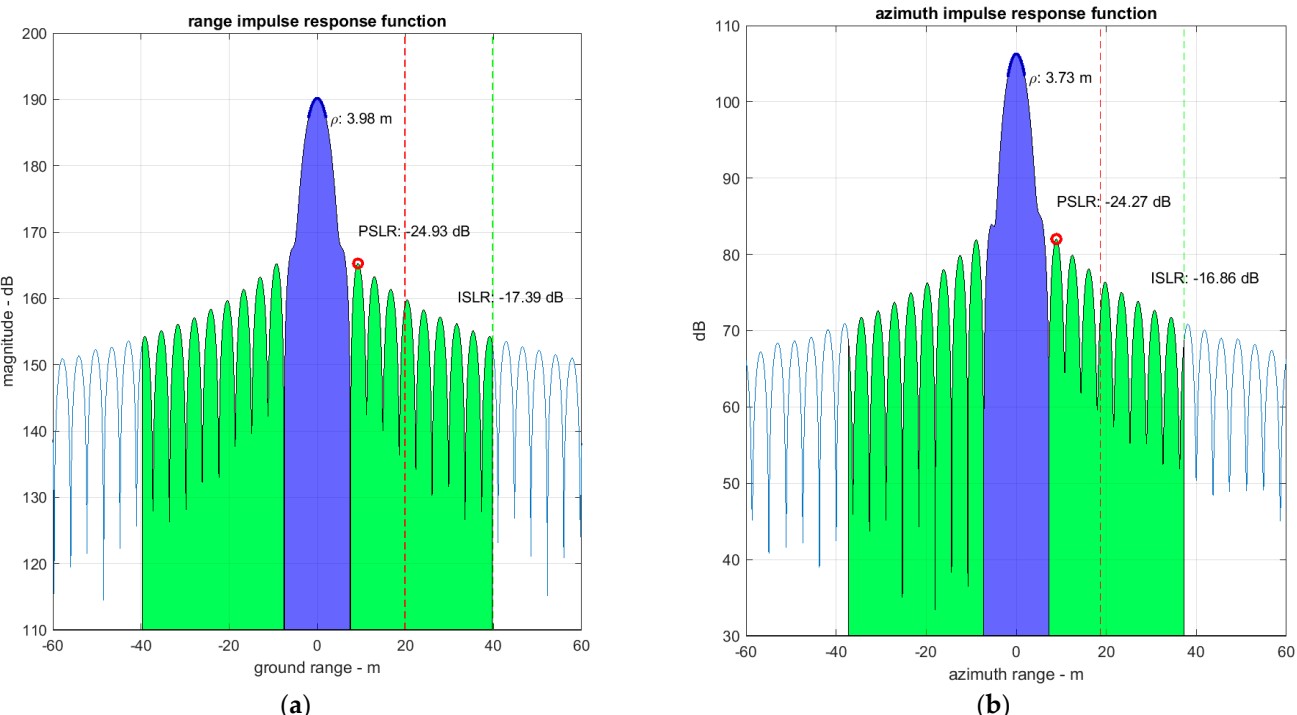

**Figure 26.** Impulse response function analysis in the (**a**) range and (**b**) azimuth of the S-STEP SAR payload in the wide-swath mode. The red circles indicate the peak side lobes.

### 6.5. Range Ambiguity Ratio

Ambiguity is measured by the ambiguity-to-signal ratio of the SAR, i.e., the ratio of the ambiguity power to the power of the useful signals [2–4]. Range ambiguity refers to echoes occurring outside the swath, and it enters the radar receiver along with the useful

echoes inside the swath, which deteriorates the radar image quality after signal processing. The range ambiguity-to-signal ratio (RASR or RAR) can be expressed as [45]

$$\mathrm{RASR} = \frac{\sum_{n=1}^{N_s} S_{a,n}}{\sum_{n=1}^{N_s} S_n}, \tag{10}$$

where $N_s$ denotes the total number of samples within the Rx echo window, $S_{a,n}$ represents the ambiguity power of the $n$-th sampling point within the Rx echo window.

$$S_{a,n} = \sum_{\substack{k=-N_p \\ k \neq 0}}^{N_p} \frac{\sigma^0(\theta_i(n,k)) G_t(\theta_i(n,k)) G_r(\theta_i(n,k))}{R_s^3(\theta_i(n,k)) \sin(\theta_i(n,k))}, \tag{11}$$

where $k$ indicates the range ambiguity number index ($k \neq 0$), $N_p$ denotes the total number of pulses within the radar horizon, $S_n$ represents the signal power of the $n$-th sampling point inside the echo receiving window, $\theta_i(n,k)$ denotes the angle of the $n$-th sampling point and $k$-th range ambiguity, and $R_s$ reflects the slant range from the antenna to the range ambiguity.

$$S_n = \frac{\sigma^0(\theta_i(n,k)) G_t(\theta_i(n,k)) G_r(\theta_i(n,k))}{R_s^3(\theta_i(n,k)) \sin(\theta_i(n,k))}, \tag{12}$$

where $\sigma^0$ denotes the backscattering coefficient of the surface clutter, $G_t$ reflects the transmit antenna pattern at elevation, $G_r$ indicates the receive antenna pattern at elevation, and $k = 0$.

The RAR was obtained using the average backscattering model for the land clutter by Ulaby [46]. The geometry and antenna gain of the range ambiguities at an incidence angle of 25° are portrayed in Figure 27a,b, respectively. The forward and backward range ambiguity points can be obtained by evaluating the slant range corresponding to the multiples of PRI selected for the specific antenna beam [47].

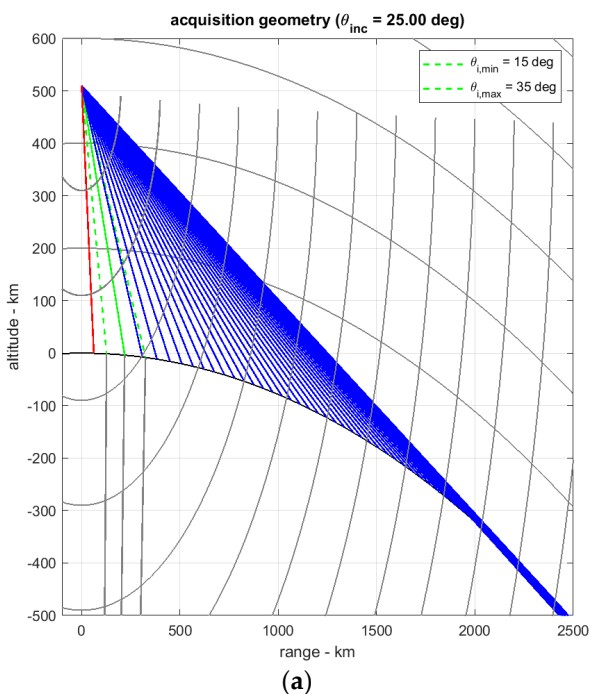

(a)

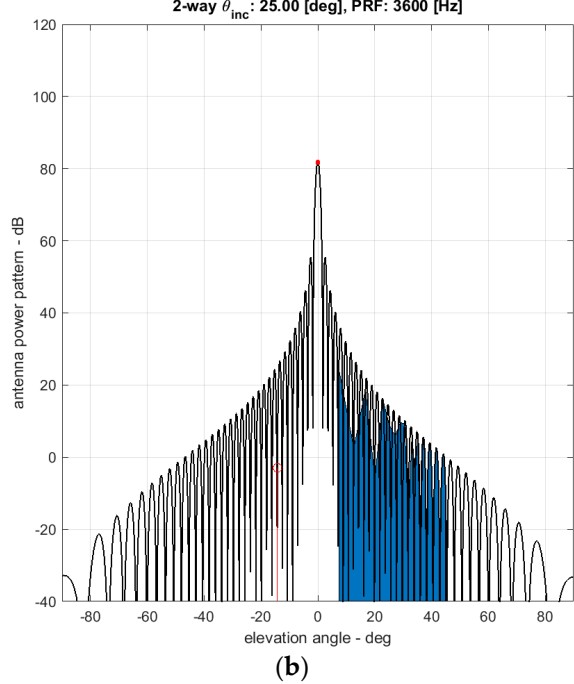

(b)

**Figure 27.** (**a**) Forward range ambiguities (blue), backward range ambiguities (red), and antenna beam boresight (green), and (**b**) range ambiguities in the elevation angle.

The RARs at incidence angles of 15 and 25° are illustrated in Figure 28a,b, respectively. As observed in Figure 28, the RAR deteriorates with the increasing PRF. The RAR according to the incidence angle in high-resolution and wide-swath modes is presented in Figure 29a,b, respectively. As observed from Figure 29, the RAR is satisfied for all operating incidence angle intervals of the selected PRFs in the high-resolution and wide-swath modes.

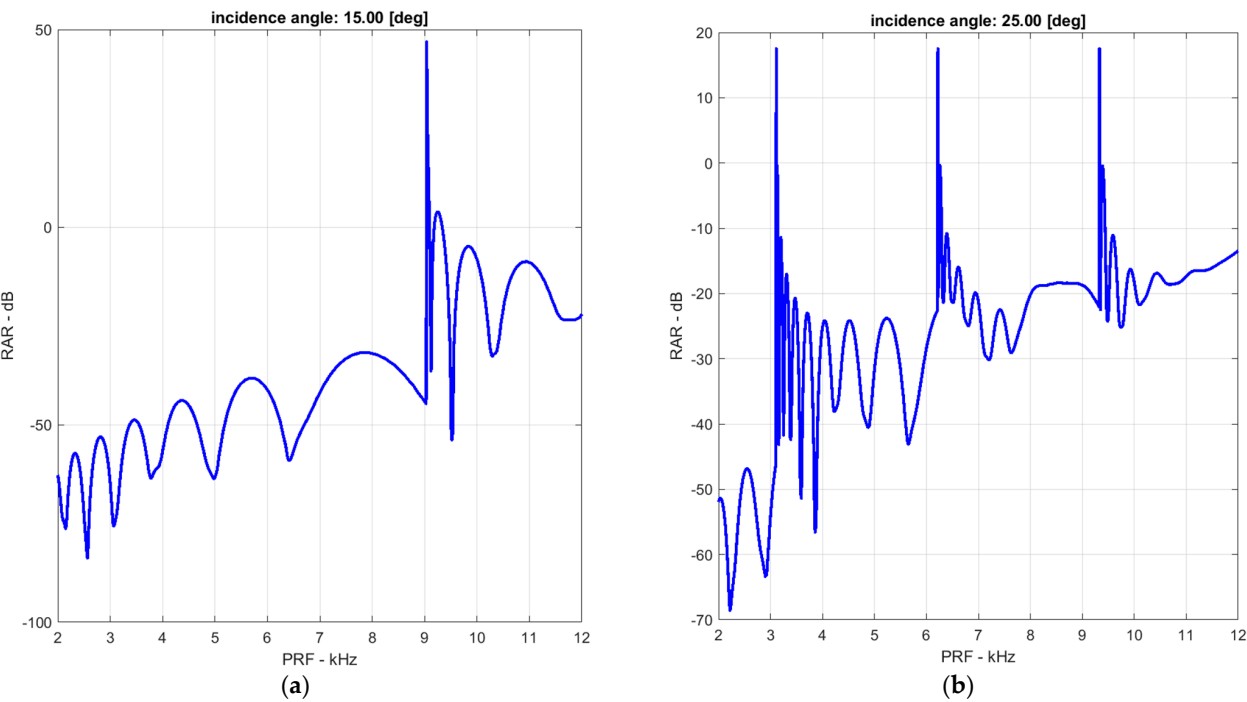

**Figure 28.** Range ambiguity ratio vs. PRF at an incidence angle of (**a**) 15 and (**b**) 25°.

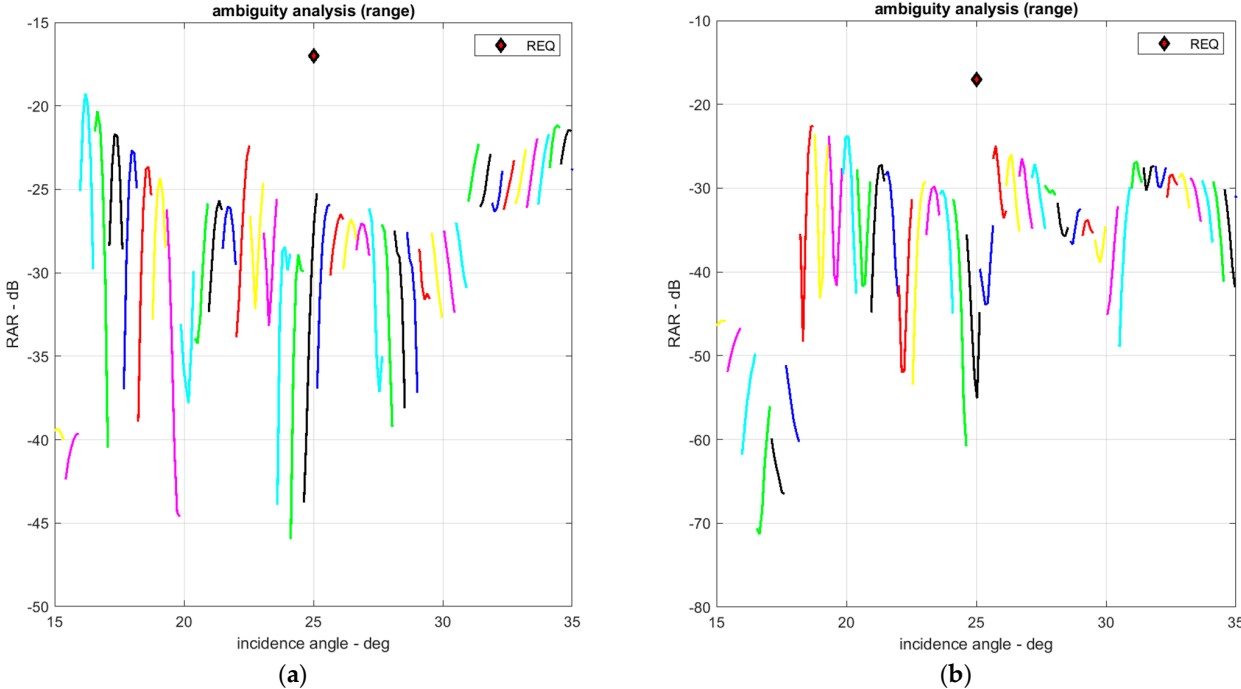

**Figure 29.** Range ambiguity ratio vs. incidence angle in (**a**) high-resolution and (**b**) wide-swath modes. RAR was lower than the required level of −17 dB at the incidence angle of 25°. Each color shows the corresponding swath.

### 6.6. Azimuth Ambiguity Ratio

The difference between the Doppler frequency of the target echoes at certain angles and the main beam is an integral multiple of the PRF azimuth ambiguity resulting from Doppler frequency aliasing. In particular, the azimuth ambiguity is primarily caused by the finite sampling of the Doppler spectrum at an interval of PRF. Moreover, the ratio of the ambiguity to the expected signal is defined as the azimuth ambiguity-to-signal ratio (AASR or AAR) [2–4], expressed as [45]

$$
\text{AASR} = \frac{\sum_{\substack{m=-M \\ m \neq 0}}^{M} \int_{-\frac{B_d}{2}}^{\frac{B_d}{2}} G_t(f_d + m \cdot f_p) G_r(f_d + m \cdot f_p) \, df_d}{\int_{-\frac{B_d}{2}}^{\frac{B_d}{2}} G_t(f_d) \cdot G_r(f_d) \, df_d},
\tag{13}
$$

where $G_t(f_d)$ and $G_r(f_d)$ denote the azimuth antenna gain patterns (one-way) for Tx and Rx in the Doppler spectral domain, respectively; $B_d$ represents the Doppler bandwidth; and $M$ denotes the maximum azimuth ambiguity number (typically $M = 30$ for performance analysis).

The AAR improves with the increasing PRF. Therefore, the RAR and AAR exhibited a trade-off relationship depending on the PRF selection. Thus, the PRF should be selected to satisfy the appropriate RAR and AAR. The antenna gains from the azimuth ambiguities in the PRFs of 3500 and 8400 Hz are presented in Figure 30a,b, respectively. Figure 31a,b imply that AAR is satisfied in all operating incidence angle intervals for the selected PRFs in the high-resolution and wide-swath modes, respectively.

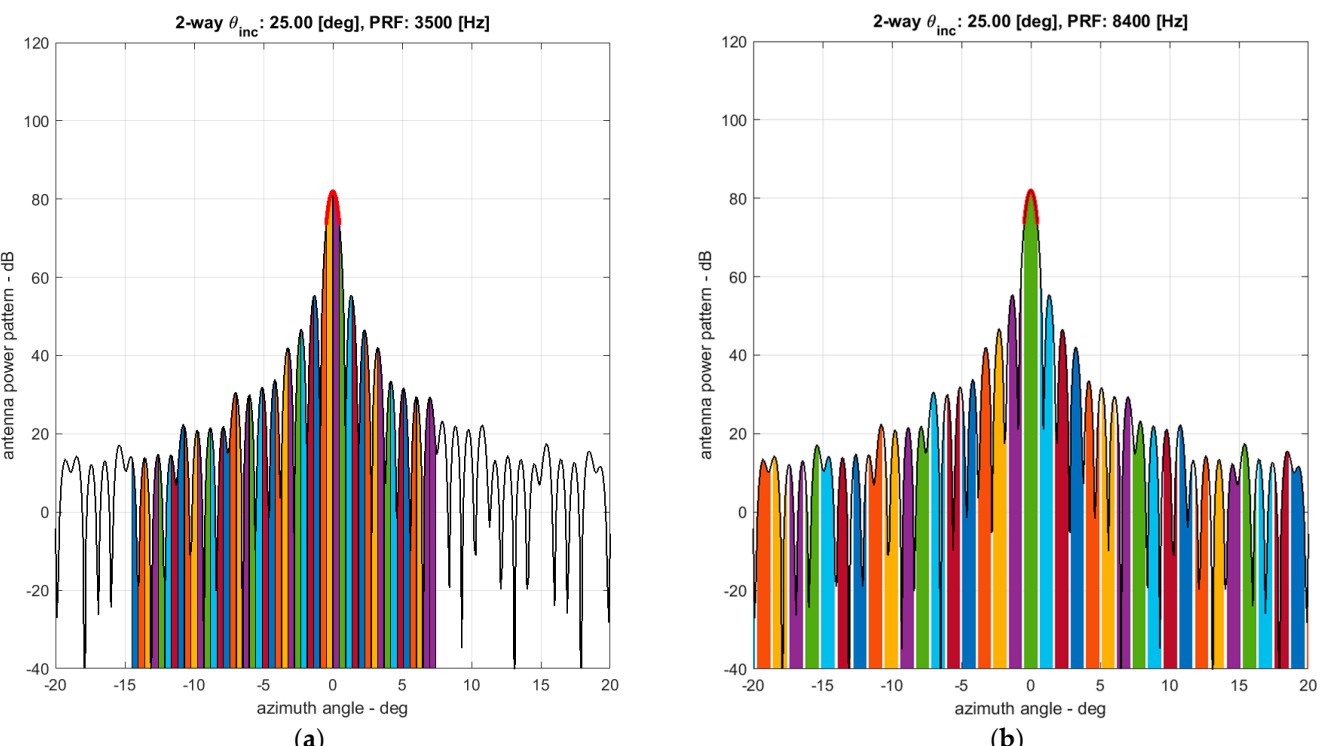

**Figure 30.** Azimuth ambiguities at PRFs of (**a**) 3500 and (**b**) 8400 Hz. Each color bar corresponds to the azimuth ambiguity at a specific azimuth angle. No overlap between azimuth ambiguities is observed for a PRF of 8400 Hz, which is larger than the Doppler bandwidth.

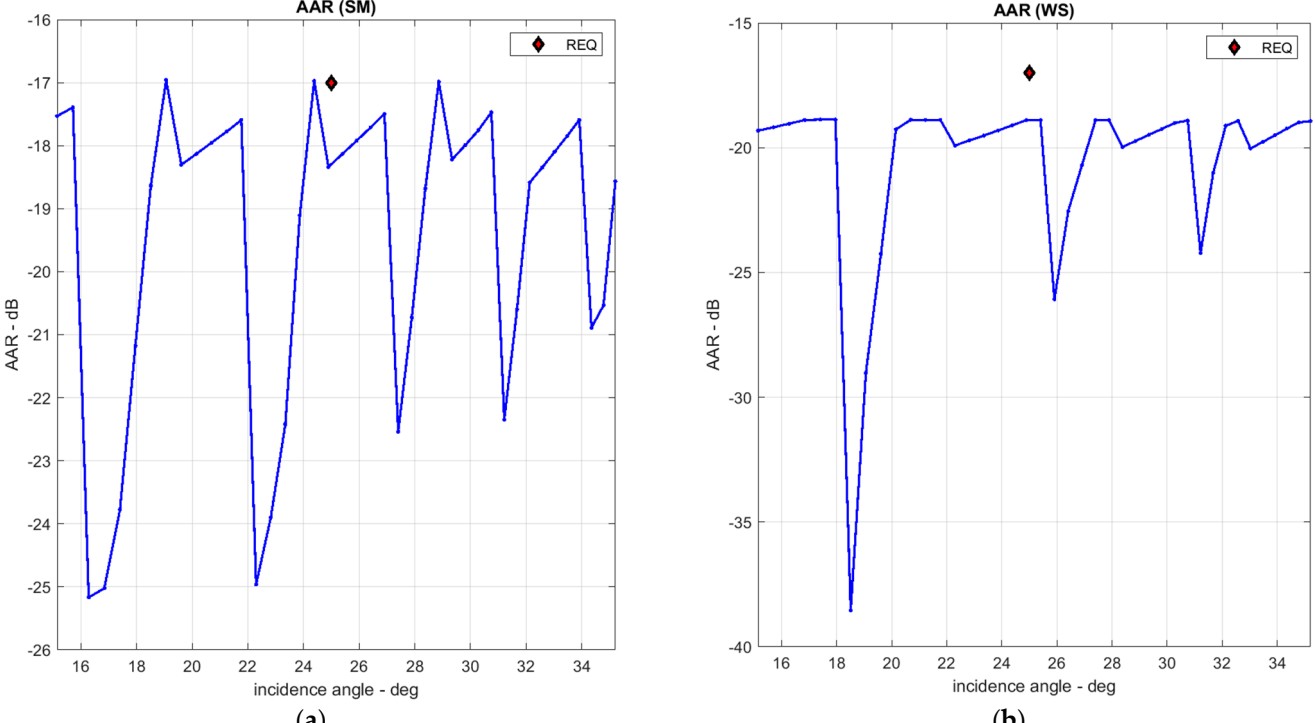

**Figure 31.** Azimuth ambiguity ratio vs. incidence angle of each antenna beam in (**a**) high-resolution and (**b**) wide-swath modes. AAR was lower than the required level of −17 dB at an incidence angle of 25°.

## 7. Conclusions

In this study, an innovative SAR payload design was proposed and investigated for the development of a new space-based 80 kg class X-band SAR microsatellite. The major design approaches, i.e., a flat-panel-type SAR payload integrated with a bus payload, were verified using an SAR performance analysis based on mathematical models. The proposed novel SAR payload can be applied to flat-panel-type microsatellites for Earth observation, communication, and internet service missions.

Additionally, a study proposed the detailed design of the SAR system and its units, which are compatible with an 80 kg class microsatellite, by focusing on the SAR payload subsystem level. If this microsatellite SAR is deployed for a typical Earth observation at an altitude of 510 km, its SAR image performance can successfully satisfy the S-STEP mission requirements. Moreover, the MBSE design results for the SAR payload were presented using the Capella MBSE open-source SW, which facilitated rapid adaptation of the varying or updating mission requirements, improvement of consistency, and improved analysis of the system.

Although the proposed design of the S-STEP SAR payload fulfilled the aims and objectives of applying an SAR on a single microsatellite of 80 kg mass, several facets of this research are required to be explored through dedicated future research. The S-STEP SAR payload will be developed through gradual performance improvements. In the future, the characteristics of the flat-panel-type shape, such as OrigamiSAT of Oxford Space Systems, Inc., can be utilized to expand and develop the S-STEP SAR into an advanced and innovative satellite concept.

**Author Contributions:** Investigations: S.K., S.-H.L. and H.-U.O.; writing—original draft preparation: S.K.; writing—reviewing and editing: S.K.; Visualization: S.K. and C.-M.S.; Supervision: S.-C.S. and H.-U.O. All authors have read and agreed to the published version of the manuscript.

**Funding:** This research was supported by the Challengeable Future Defense Technology Research and Development Program (912777601-9127776-04) of Agency for Defense Development (ADD) in 2022.

**Conflicts of Interest:** The authors declare no conflict of interest.

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
