# Peer review of "Design and Performance of X-Band SAR Payload for 80 kg Class Flat-Panel-Type Microsatellite Based on Active Phased Array Antenna"

_aerospace, doi:10.3390/aerospace9040213_

Round 1
Reviewer 1 Report
Micro SAR satellite is a research and application trend in the field of SAR satellite. This paper gives the detailed design process and demonstration process of key parameters of micro SAR satellite. In general, this is a very meaningful research work.
I have some suggestions to share with the author.
- Only single polarization (VV) is designed for the SAR payload. From the perspective of application requirements, multiple polarization is better, such as Full polarization, Compact polarization or Dual polarization. For planar phased array antennas, it is possible.
- The incident angle is 15-35, the max 45 may be considered. However, the NESZ will be degraded. This is just a suggestion.
- The NESZ with -14 dB may be not adequate for SAR image.
Author Response
Response to Reviewer 1 Comments
Thank you for your valuable review comments and suggestions.
Point 1: Only single polarization (VV) is designed for the SAR payload. From the perspective of application requirements, multiple polarization is better, such as Full polarization, Compact polarization or Dual polarization. For planar phased array antennas, it is possible.
Response 1: As you mentioned, the polarization diversity is possible. But there are several consideration and constraints. This proposed SAR micro-satellite is used for the satellite constellation of more than 30 satellites. According to the system design result at the satellite level, the polarization diversity will be applied for satellite constellation level, which means one satellite has vertical polarization and another has horizontal polarization considering the simplicity and cost reduction at this development stage. We will look on the possibility of dual polarization on a single SAR antenna without severe cost increase in terms of the performance improvement.
Point 2: The incident angle is 15-35, the max 45 may be considered. However, the NESZ will be degraded. This is just a suggestion.
Response 2: As you said, in general, we could consider increasing the operational incidence angle up to above 45 degrees. But specifically in case of micro-satellite with less than 100 kg, the transmitted power is usually very limited in terms of mass, power consumption, and heat. More transmitted power is needed in order to increase the incidence angle. So the existing SAR micro-satellites are usually operated within this incidence angle interval of 15 to 35 degrees. At the beginning stage of the development, we will set this incidence angle as a baseline. But we will look on this point for the improvement.
Point 3: The NESZ with -14 dB may be not adequate for SAR image.
Response 3: The less NESZ, the better sensitivity of the SAR image quality. This NESZ requirement is the best acceptable level at the beginning stage of the development, within the limitation of mass, power, size and cost. As we know, this level of NESZ is acceptable for general application. We will look on the performance improvement after the hardware development and test.
------------------------------------------------------------------------------------
We will partly revise the manuscript considering the reviewer review comments.
Reviewer 2 Report
Original Submission
Recommendation
Minor(or somehow accepted) revision
Comments to Author:
Title:
Design and Performance of X-band SAR Payload for 80-kg class Flat-Panel-Type Microsatellite based on Active Phased Array Antenna
Overview and general recommendation.
This paper is demonstrating usage of SAR data and some important applications. This technique is very important and somehow accurate and has lots of important applications. First of all, as a person with more than 20 years familiarity with SAR/RADAR data, I like this paper very much; but as a scientist, I have to say the truth about the material and to be honest.
The Abstract is OK: summarize the idea and concepts inside the paper; English is a big problem!! I think it is better to give it to a native person to review (it is a big must).
Introduction is fair. I think in some positions, some important corrections must be done; leak of some Refs... please fix them! Pls improve the quality of some Figs => they are very bad; unacceptable (I do not accept this!).
I like this paper very much: good experiments have been done; however, I think this work must be improved and lots of things to do; agreed? Pls do these primary corrections, then I will go over the paper again...
Detailed comments:
Fig2, 3… Quality? All of them…
Author Response
Response to Reviewer 2 Comments
Thank you for your valuable review comments and suggestions.
Point 1: The Abstract is OK: summarize the idea and concepts inside the paper; English is a big problem!! I think it is better to give it to a native person to review (it is a big must).
Response 1: As you mentioned, we will improve and correct the manuscript with a native professional speaker as much as possible.
Point 2: Introduction is fair. I think in some positions, some important corrections must be done; leak of some Refs... please fix them! Pls improve the quality of some Figs => they are very bad; unacceptable (I do not accept this!).
Response 2: Thank you for your comments. We will check and correct the references in the manuscript. Also, the resolution of all the figures in the manuscript will be improved to more than 300 dpi.
Point 3: I like this paper very much: good experiments have been done; however, I think this work must be improved and lots of things to do; agreed? Pls do these primary corrections, then I will go over the paper again...
Response 3: The manuscript currently focuses on the design of SAR payload at this design phase. We present the numerical verification results and analysis results of the proposed SAR payload design in the manuscript. We will improve and revise partly the verification results and analysis results of the manuscript in more clear way. We are manufacturing the SAR payload equipments. We will prepare and verify the test results of the SAR payload at equipment and payload level later as the development goes on in the future after the test phase.
----------------------------
We will revise and improve the manuscript according to the valuable reviewer comments and suggestions, including the English editing.